# Polymer-Matrix Composites: Characterising the Impact of Environmental Factors on Their Lifetime

**DOI:** 10.3390/ma16113913

**Published:** 2023-05-23

**Authors:** Rui Barreira-Pinto, Rodrigo Carneiro, Mário Miranda, Rui Miranda Guedes

**Affiliations:** 1Departamento de Engenharia Mecânica Faculdade de Engenharia, Universidade do Porto, Rua Dr. Roberto Frias, 4200-465 Porto, Portugal; ruibmpinto@gmail.com (R.B.-P.); up201806895@edu.fe.up.pt (R.C.); up201806576@edu.fe.up.pt (M.M.); 2INEGI-Instituto de Engenharia Mecânica e Gestão Industrial, Rua Dr. Roberto Frias, 4200-465 Porto, Portugal

**Keywords:** polymer-matrix composites, environmental fatigue, creep, water absorption, seawater, lightning strike damage, corrosive media, UV damage, bacterial degradation

## Abstract

Polymer-matrix composites are widely used in engineering applications. Yet, environmental factors impact their macroscale fatigue and creep performances significantly, owing to several mechanisms acting at the microstructure level. Herein, we analyse the effects of water uptake that are responsible for swelling and, over time and in enough quantity, for hydrolysis. Seawater, due to a combination of high salinity and pressures, low temperature and biotic media present, also contributes to the acceleration of fatigue and creep damage. Similarly, other liquid corrosive agents penetrate into cracks induced by cyclic loading and cause dissolution of the resin and breakage of interfacial bonds. UV radiation either increases the crosslinking density or scissions chains, embrittling the surface layer of a given matrix. Temperature cycles close to the glass transition damage the fibre–matrix interface, promoting microcracking and hindering fatigue and creep performance. The microbial and enzymatic degradation of biopolymers is also studied, with the former responsible for metabolising specific matrices and changing their microstructure and/or chemical composition. The impact of these environmental factors is detailed for epoxy, vinyl ester and polyester (thermoset); polypropylene, polyamide and poly etheretherketone (thermoplastic); and for poly lactic acid, thermoplastic starch and polyhydroxyalkanoates (biopolymers). Overall, the environmental factors mentioned hamper the fatigue and creep performances, altering the mechanical properties of the composite or causing stress concentrations through microcracks, promoting earlier failure. Future studies should focus on other matrices beyond epoxy as well as on the development of standardised testing methods.

## 1. Introduction

Polymer composites have long been a focus of intensive research, especially since they are, now more than ever, regarded as viable alternatives for many engineering applications. However, when alone, their properties, particularly mechanical strength and stiffness, do not usually completely satisfy the requirements of a given application, hence the need for reinforcements and the advent of polymer-matrix composites (PMCs). These reinforcements can be of multiple shapes, ranging from dispersed particles to fibres aligned along a preferential direction, effectively altering the behaviour of the material as a whole. They are all bonded together and held in place by a polymeric medium, i.e., a polymer matrix, responsible for transferring and distributing external loads in the composite and for protecting the reinforcements from the elements.

Despite their importance in withstanding mechanical loads, reinforcements barely account for any weathering and degradation protection, which falls entirely upon the matrix, nor do they dictate the fatigue and creep life of a composite. Hence, throughout this review, we follow the established conclusion that the fatigue and creep behaviours depend on the matrix, and focus on the latter, wherein damage usually initiates by cracking, due to their generally poor resistance to corrosion and inferior mechanical properties.

Matrices can be either thermoset or thermoplastic, with the former including the vast majority of PMCs suited for structural applications, whereas the latter, whilst not so stiff and versatile, have seen a recent development driven by sustainability and recycling concerns. Additionally, renewable organic matrices, even if still thermoplastic, constitute on their own a class of materials with growing importance, predominantly in biological and medical applications, where they also experience fatigue and creep. Before we move on, however, we note that, while pure mechanical fatigue in polymer-matrix composites is typically cycle-dependent, environmental effects introduce time-dependency, as they alone suffice to degrade a given polymer given enough time [1].

Accordingly, this review verses on the influence of environmental effects on the fatigue and creep behaviour of a PMC, phenomena that, contrarily to the case of metals, are still not completely understood and are still a topic of dispute and prolific research among the scientific community. It aims at not only describing how several different environmental conditions interact with the micro-structure of a polymer but also at listing comprehensively the most important results found in the literature for typical polymer matrices.

In Part I, we describe the environmental effects considered and how they affect polymers and their microstructure. Respectively, we focus on solvent absorption—especially water and moisture, seawater, liquid chemical agents, microorganisms, UV/ionising radiation, temperature variations—from cryogenic to high-temperature realms—and lightning strikes.

In Part II, we selected for each group of polymer matrices (thermoset, thermoplastic and biopolymers), three representative polymers, the most often found or the most promising for engineering applications, namely, epoxy, vinyl ester and polyester (PE) for the thermoset group; polypropylene (PP), polyamide (PA) and polyetheretherketone (PEEK) representing thermoplastics; and polylactic acid (PLA), polyhydroxyalkanoate (PHA/PHB) and thermoplastic starch (TPS) for the biodegradable and bio-based matrices. We, then, compile the available literature for each polymer mentioned, explaining the impact of environmental factors on different composites and elaborating, whenever pertinent, on the particle-/fibre–matrix interactions.

At this point, an important remark must be mentioned; albeit we focus on environmental creep and fatigue and not on degradation, we still mention degradation mechanisms and effects to provide a better understanding of the phenomena involved, always keeping in mind the scope of this work. This is especially true in the case of biopolymer-matrix composites, for which very few works have been conducted on mechanical behaviour, but many more exist on the degradation effects from a chemical standpoint. Thus, insight into creep and fatigue and their likely trends can still be gathered from the available literature, never doing away with further experimental studies to investigate these phenomena.

Finally, we present our final remarks and reflect on the literature reviewed, pointing out areas where very little information is available that future research should tackle.
**Part I—Environmental factors**

## 2. Solvents

Contrarily to ceramics or metals, polymers may accommodate different organic or inorganic solvent molecules in their microstructure, their solubility depending on the solvent of interest: polar solvents, as a general rule, permeate polar polymers and nonpolar solvents permeate nonpolar polymers [2].

Bearing in mind that the moduli of polymers are controlled by (1) binding forces (particularly van der Waals forces), responsible for the elastic behaviour, and by (2) the unfreezing of molecular mobilities (dispersion steps), accounting for viscoelastic effects, we anticipate the incorporation of a solvent that alters this temperature-dependent equilibrium of forces. When the molecules of the solvent enter the polymer medium, they increase the distance between the chains, occupying the vacant spaces in between them, which is accompanied by a volume increase in the polymer material itself. This swelling (Figure 1) phenomenon weakens the intermolecular bonds, as they are now stretched with respect to their dry configuration, with consequences to the thermomechanical properties of the polymer, e.g., above the glass transition temperature, $T_{g}$, swelling reduces the strength of a polymer and/or introduces residual stresses if the solvent uptake is non-uniform within a sample [2].

On a different line, solvents, by weakening the intermolecular bonds, may assist in the crazing and fracture phenomena as well, since they not only reduce the surface energy required to create and propagate a free surface, but also diminish the bond strength and, consequently, the force needed to pull a chain out of the bulk, facilitating the formation of fibrils and, thereby, of cracks. Thus, some polymers may become brittle by incorporating a solvent [2], potentially resulting in a decrease in their fatigue life.

### 2.1. Water

Being the most important and widely available solvent, water (polar) is, then, of particular interest. Not only can it bind itself to strong polar groups by hydrogen bridges [4], as, additionally, in high-enough quantity and over an extended period, its presence may lead to the hydrolysis of the polymer, i.e., the cleavage of bonds in long chains, due to the presence of water molecules, into short monomer-like groups whose properties differ from those of the original polymer. This phenomenon affects especially polymer matrices capable of absorbing significant quantities of water/moisture, such as polyamides, hindering their application in particularly humid and wet environments [2].

Delving into more detail, the uptake of water shifts the glass transition temperature of a polymer to lower values, with further effects depending on the actual working temperature; as such, we distinguish two realms: low temperatures ($T<T_{g}$) and high temperatures ($T≳T_{g}$, which generally includes room temperature). In the former domain, especially at cryogenic temperatures, since the mobilities of segments are suppressed and the moduli are controlled mainly by the van der Waals forces, we observe, first, that wet specimens are stiffer than dry ones, i.e., higher Young’s and shear moduli—stiffening effect, as frozen water stiffens the polymer chains it is linked to. Secondly, we note that the damping spectra and loss factor, δ, are shifted by the presence of water [4], which is related to the unfreezing behaviour, since a damping peak—an abrupt increase in loss factor—indicates the onset of unfreezing, which is affected by the volume of water present, with the main increase, and maximum loss factor, corresponding to $T_{g}$.

Contrarily, since, at $T_{g}$, the mobilities of segments unfreeze and the chains can now flex and move more freely, for temperatures close to or above the glass transition, water is considered to act as a softening/plasticising agent, i.e., a spacer between chains according to the mechanisms detailed earlier, reducing the static strength, stiffness and $T_{g}$ of a given polymer proportionally to the change in the loss factor, while simultaneously increasing its fracture strain and impact strength [1,4]. This is predominantly seen in the amorphous regions of a polymer, which contribute more to swelling than the crystalline ones, where water sorption occurs solely in the crystallite surfaces, i.e., the higher the crystallinity degree of a polymer matrix, the lower its swelling capacity [1,5], which is further confirmed by the observation that thermoset (amorphous) matrix composites feature saturation limits of 1–2% and thermoplastic (amorphous and semi-crystalline) matrices around 0.1–0.3% [1].

Nevertheless, even if most polymers absorb water, not only does its quantity vary, but also the degree to which the properties of a polymer change—if they change—depends on a plethora of factors, such as temperature, time, thickness of specimen, and the amorphous content in a polymer. For some polymers, even a low water content might induce a large effect.

Focusing specifically on fatigue and creep behaviour, we note that hydrolysis, with the subsequent plasticisation and decrease in matrix-related properties (shear, transverse tension and longitudinal compression), generally increases the creep and stress relaxation rate and the likelihood of environmental stress cracking of a composite [1]. Furthermore, the swelling/shrinking cycles in composites create voids between the matrix and the reinforcements (Figure 2), due to the different expansion coefficients—that of the matrix being generally larger than that of the reinforcement, which generate internal stresses and damage the interface region, as fillers begin to occupy the voids, increasing the tension therein in the subsequent cycles. After enough cycles, fully formed gaps appear, leading to cracks on the inside of the material that extend outwards over time [5]. This extension can be understood, since, during the loading stage, cracks in the matrix or in the fibres become infiltrated with water, and, during unloading, they cannot fully close, compressing the liquid within them. Thus, considering the almost total incompressibility of liquid water, large pressures act cyclically in the microcracks and in the flooded capillaries, increasing the amount of damage the composite experiences (fibre—resin debonding, matrix cracking and swelling) and reducing its fatigue life [1,6,7,8]. Nevertheless, this occurs solely above a given water threshold, with studies indicating lower lifetimes at high strains and longer lifetimes at low strain levels for wet samples compared to dry ones, attributing this behaviour to the plasticising effect, since the humid matrix is less brittle than the dry one and, thus, more prone to microbuckling at high strains [9]. Contrarily, this is offset by the lubrication effect of water that reduces the internal friction between crack faces, lowering the levels of resistance to fracture for all strain levels [8]. It follows that the overall wet fatigue response is a competition between these two phenomena, hence the difficulty in describing it physically.

### 2.2. Seawater

Although one might be tempted to include seawater effects under the previous subsection and while it is true that some effects are common as they derive solely from the water uptake, there is a plethora of additional phenomena that must be accounted for when analysing the effect of seawater in polymer matrices.

First, in comparison to other environments, the marine environment is defined by conditions as diverse as high salinity (seawater is rich in inorganic salts), high pressure, currents, and low nutrient levels (e.g., nitrate) [11]. Second, the water temperature, besides depending on the geographical region and on the season, diminishes with the increase in depth: at the surface, the average temperature is 17.4 °C, dropping to 0–4 °C at a depth of more than 2000 m. Likewise, the salinity follows the same trend, changing with season and area, the surface average of the ocean being 35%, while that of offshore and estuary water sits generally below 30%. It follows that seawater has then a slightly alkaline behaviour, with middle and deep seawater having a pH value between 7.5 and 7.8, and at the surface, this value increases to 8.1 ± 0.2 [11]. Naturally, all these factors coupled with both the high ionic strength of seawater and the inherent buffer capacity may well exacerbate the degradation driving force when compared to neutral water.

The abiotic factors just mentioned influence the biological habitats, causing them to be vertically structured, with not only varying temperature, salinity and pH values, but also pressure, light and UV radiation levels; below 4000 m, it is completely dark and cold and the pressures are immense, leading to a stratified distribution of marine species, whose biological processes, in turn, affect the dissolved oxygen content in seawater [11], which further adds to the factors impacting the performance of a PMC.

More specifically, the strong UV light may favour crosslinking, as explained in Section 5, embrittling a polymer, and the mechanical forces due to currents and waves may help break down large specimens into smaller pieces, thus, favouring hydrolysis, both abiotic and biotic. The former occurs at all times, even if slower due to the low temperatures of the medium, whereas the latter is particularly relevant for biodegradable materials, which, as the name entails, are affected by the existing microorganisms. Again, decomposition occurs slower in this case than it would in other environments, e.g., in soils or compost, owing to the lower amounts of active microbial communities capable of withstanding the low temperatures and high salinity. A decreased biodegradation rate in seawater has been observed in the literature for different polymer matrices [11,12,13]. Then, it should not come as a surprise that even biodegradable materials with rapid degradation kinetics on the waterfront may not degrade for a long time after entering the ocean or sinking into the deep sea, contributing to ocean pollution.

Concerning its influence on fatigue, similar results to those found for pure water have been reported [6,7,8,9], with the water molecules increasing the stress levels within the microcracks upon cyclic loads, although hydrolysis takes longer to occur, since seawater is typically colder that room temperature water. In addition, the biotic factors must still be accounted for, as the presence of microcracks increases the area exposed to bacterial or fungal attack, resulting in a faster loss of mechanical properties for some susceptible polymers, e.g., PHB.

## 3. Liquid Chemical Agents

Owing to their notable corrosion resistance, PMCs are replacing their more conventional metallic counterparts in aggressive corrosive—chemically active—environments and applications, e.g., containment or coating functions to protect a structure from a variety of chemicals [14,15,16]. On the one hand, physical aspects of degradation do not induce changes in the chemical structure and may be reversible [10,16,17,18]; on the other hand, chemical deterioration is defined as the set of mechanisms altering the chemical structure of the material [16], resulting from exposure, e.g., to acidic or alkaline solutions [19].

In contrast to metals, whose corrosion behaviour is given by an electrochemical reaction between the material and the surrounding chemical medium occurring predominantly at its surface [16,20], polymeric materials are in general insulators and, thus, do not dissolve in the form of ions [17,19,20]. Moreover, corrosion in polymers occurs within a reaction area, ranging from a few monolayers to its complete bulk [16], and infiltration of surrounding agents into the polymer may also occur, affecting the extent of corrosion within the material [20]. This may include:Depolymerisation—extraction of the monomers from the end of the macromolecules;Decomposition of the monomers—cleavage of a set of atoms from the macromolecules whilst preserving the original degree of polymerisation;Chemical breakage of the macromolecular chains leading to a decrease in the degree of polymerisation;Cross-linking processes that result in the creation of branched and network architectures by creating chemical links between the macromolecules [16,20].

Corrosion and performance decline of the polymer is the result of such chemical reactions that damage the primary bonds [20]. The chemical solubility of a PMC is largely determined by the surrounding polymer matrix, and the core aspect of resistance to such processes is the type of active groups—chemical nature—that the matrix possesses [14]. The likelihood of damage increases with the chemical affinity of the solvent to the structure of the matrix, e.g., polymers composed by ester, amid, ether or silicon–oxygen bonds easily hydrolyse in acidic or alkaline environments [14,20].

From the concepts of diffusion and reaction rate, three types of corrosion are understood, namely surface reaction, corroded layer forming or penetration. Corrosion behaviour follows a surface reaction or corroded layer forming type when the diffusion rate is considerably lower than the reaction rate. The former emerges when the reaction rate is so rapid that the corrosion spreads uniformly across the entire surface, and thus, the dissolution initiates precisely where the liquid contacts the surface. The cured resin breaks down at cross-linking locations in the main chain, in turn producing low molecular weight products. As for the latter, a residual corroded layer with the original bonding to uncorroded resin is generated as fragments of the corroded resin dissolve out. Even though bonds are broken by hydrolysis and the layer becomes eroded, the molecular chains are still held together, and hence, the polymer exhibits a high molecular weight. In addition, the corroded layer behaves as a barrier to solution penetration, and accordingly, the reaction occurs at the interface between the layers. In Figure 3, we observe that the corroded layer might be coarse (case 1) or tight (case 2) as a function of the chemical structure of the polymer [19].

On the other hand, penetration takes place when the diffusion rate exceeds the reaction rate. This type of corrosion behaviour is characteristic of polymers and is defined by an initial penetration of the solution into the resin until an equilibrium stage is reached, followed by a strength reduction owing to the chemical reactions that degrade the polymer [19]. Such a process is not only critical for the corrosion behaviour but also for the environmental fatigue and creep performance of PMCs under liquid chemical mediums. In virtue of defects introduced during manufacturing and cracks that develop under cyclic loading, the surrounding environment is able to penetrate the PMC. Initially, diffusion of the solution into the matrix results in swelling—reversible first stage of physical damage. With increasing exposure time, however, the fluid penetrates deeper within the composite into larger areas containing defects—where the micro-cracks and cracks converge. This provides enough space for chemical reactions to occur, and when exposed to acidic solutions, reactions between the H${}^{+}$ ions and the functional groups of a vinyl ester resin causing its dissolution were recorded [18]. Such a process is naturally irreversible and additionally decreases the compactness of the matrix, ultimately reducing its ability to uniformly transfer the load between the fibres, incurring in mechanical performance and fatigue life losses [18].

Furthermore, although one of the primary functions of the matrix is to protect the fibre from the surrounding environment and the corrosion behaviour is initially dependent on the resistance of the resin, the interface and fibre also play a critical role in providing chemical resistance and fatigue and creep performance [14,19]. Augmenting the aforementioned exposure period even further leads to propagation of the solution to the bonding interface between the resin and fibres (Figure 4), whose coupling agent is usually amphiphilic with one hydrophilic end and the other hydrophobic. Due to the capillary effect of the interface, fluid permeation results in a shear effect between the resin and interface which is followed by chemical reactions that destroy the bond between them [18]. This was once more identified in basalt/vinyl ester specimens where hydrogen ions in an acidic solution reacted with the coupling agent, in virtue of its hydrophilic nature [18]. Similar conclusions were also reported for E-glass/PE composites [21] with the extensive interface damage displayed in Figure 5. Naturally, as the interface plays a key role in the properties of the PMC, such destruction results in deterioration of the mechanical performance [18,22].

Ultimately, given sufficient exposure time, a point may be reached where the fibres themselves are exposed to the corrosive medium. This is critical when dealing, e.g., with glass fibres that are known to severely deteriorate when exposed to acids [22,23] as a consequence of non-siliceous ions in the glass being replaced by the hydrogen ones in the acid—leaching. This may also generate axial and spiral cracks (Figure 6) on the fibre surface that lead to strength reduction [23].

## 4. Biotic Factors

Although one might at first exclude microorganisms from environmental effects that affect the fatigue and creep behaviour of a PMC, their relevance grows if we consider bio-based and biodegradable matrices. We can, thus, distinguish two different types of environmental factors, namely, abiotic ones (temperature, pH moisture, sunlight intensity, humidity, salinity and abrasion) and biotic ones, which include the density and diversity of the bacterial and fungal populations in a particular environment [25]. As already mentioned when describing seawater effects, polymeric matrices can be sources of nutrients for the microorganisms of a given environment, which decompose the former via a three-stage process: (1) fragmentation, (2) hydrolysis and (3) assimilation [11]. First, polymers are broken down to smaller pieces; then, in stage (2), as the name hints, these fragments are hydrolysed to their basic monomers, suffering a reduction in molar mass, which allows them to be rapidly metabolised (bioassimiliation and mineralisation) via enzymatic action in stage (3), i.e., broken down into cell biomass, CO${}_{2}$ and H${}_{2}$O, aerobic decomposition, or biomass, CO${}_{2}$, H${}_{2}$O and methane for anaerobic decomposition [11,12]. Two different processes can be distinguished, namely direct action—whereby the polymers themselves constitute a direct nutrient for the microorganisms—and indirect action—when the metabolic products of the microorganisms induce discolouration and/or deterioration. Both processes are influenced by the moisture level, temperature, pH of the medium and nutrient supply [12].

Regardless of the type of the process, contact occurs at the surface of a composite, with a rough surface (e.g., from mechanical break-down) favouring chemical biodegradation much more than a smooth, inert and hydrophobic one. In addition, shall there be any instances of delamination between reinforcements and the matrix, contact may initiate due to the penetration of biological agents at the interface, which may then diffuse through the matrix, particularly in amorphous domains, as semi-crystalline ones are not so easily biodegradable [5]. As crystalline regions feature ordered polymer chains, their organised structure limits the availability and accessibility of sorption sites, slowing their decomposition.

The rate of degradation, measured by weight loss, gas evolution or by accelerated respirometric tests [12], is usually bottlenecked by stage (2) defined above, with biotic hydrolysis being 8–20 orders of magnitude faster than abiotic one [11]. Focusing on the former, a plethora of factors must be considered, as it varies from environment to environment and from composite to composite, going beyond the matrix, since some reinforcements also play a significant role, e.g., natural fillers increase permeation sites. Further examples include different media, e.g., clay soil is know to accommodate a larger number of microbial communities than sandy soils and seawater, whose low temperatures do not allow for dense biota [25]. Moreover, even the same type of soil can boast different microorganisms, as most are stratified vertically, with fragments of composites being successively attacked by the different layers [25]. We must also mention composting conditions herein, which generally combine temperature and humidity values and specific biota optimised to decompose in the shortest possible time typical common use polymers and their composites.

Finally, regarding fatigue and creep behaviour, cyclic loads, especially tensile ones, increase the area available for microbial penetration, not only stretching the composite, but also opening its surface microcracks, promoting the further attack of bacteria and fungi. High-cycle fatigue loads can, however, induce a retarding effect, hindering the spread of microorganisms which may not resist the autogenic local heating resulting therefrom.

## 5. UV Radiation

When looking at the environmental fatigue of PMCs, another important factor to consider is *photo-ageing*, particularly UV ageing—under solar radiation, mostly UV-B (315–280 nm) [5]—or artificial light. This type of radiation is highly energetic, and thus, it may degrade the mechanical properties of a PMC over time, namely by photolysis, photo-catalysis and/or photo-oxidation, the latter being the most prevalent one [26]. Naturally, since the UV photons must first be absorbed, if the matrix of a PMC is impervious to this radiation, no damage will occur [1].

Damage occurs typically by chain scission and/or chain crosslinking (Figure 7). In the former, the chemical bonds within the molecular chains are broken, reducing their length and the molecular weight—as radicals are formed—and, ultimately, diminishing the glass transition temperature of the polymer [2,27]. The shorter chains are not only more mobile but are also able to crystallise easily, which results in an increase in crystallisation and, in turn, in the associated embrittlement of the matrix over time [1]. This mechanism affects preferentially the amorphous phase of a polymer. Chain crosslinking, however, since it diminishes the molecular mobility, linking chains as the name hints, embrittles directly the polymer matrix, contributing down the line to the formation of (micro) cracks and taking a toll on the fatigue life [2,27]. It affects imperfect crystalline regions but does not change the PMC chemistry [1]. Both mechanisms—chain-scission and crosslinking—can occur concurrently, as, e.g., in polyethylene matrix composites [27].

At the continuum scale, for PMCs, UV degradation is confined to the surface of the material, where discolouration and chalking, i.e., signs of oxidation, initiate usually only down to a depth of 10 μm depth (Figure 7), which corresponds to the thickness of resin at which the intensity of the UV radiation is reduced to one half of that hitting the surface [1]. In bulk polymers, nevertheless, it is not limited to this thin layer, instead accumulating thermomechanical stresses which extend gradually into the bulk, usually reaching about 0.5 mm [5], leading in the long run to creep strains, cracks and loss of tensile and impact strength [1,27].

Additionally, one must account for the interaction between factors, such as the combination of air, as an oxidising agent, with solar radiation, which accelerates the surface corrosion [5], or even the dissociation of radicals from the highly reactive broken chains that may react, e.g., with water or oxygen in the air, to form corrosive environments, such as that of PVC into chlorine radicals which form hydrochloric acid (HCl), further embrittling the composite, reducing its mechanical strength and accelerating the removal of material [1,26].

This is especially important when assessing the fatigue and creep performance, as the embrittlement causes the matrix to break earlier on when loaded and, most importantly, affects the interface between reinforcement (especially for long fibres) and matrix, as the long chains therein break, propagating the crack through the several layers of the composite and reducing the anticipated failure stress [28]. Furthermore, as the newly developed microcracks open when loaded in tension, UV radiation penetrates deeper and embrittles their exposed faces, which under large deformations, even compressive ones, can result in further cracking, again leading to premature failure.

**Figure 7 materials-16-03913-f007:**
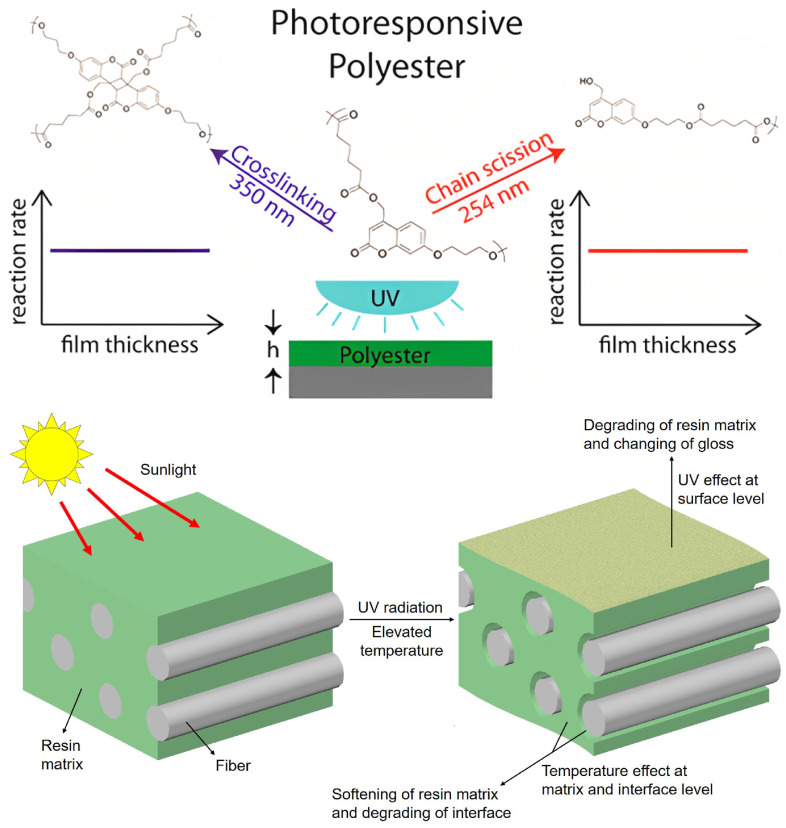
Sunlight and UV-light effects on polymer matrices. Reprinted with permission from Liu et al. [10]. Copyright 2020 Elsevier. Reprinted with permission from Lee et al. [29]. Copyright 2014 American Chemical Society.

## 6. Temperature

Similarly to metals, temperature is a crucial factor when it comes to the design and determination of mechanical, chemical and physical properties of polymer-matrix composites [30]. Due to the widely known degradation of mechanical properties of polymers with the increase in temperature, especially close to the glass transition temperature $T_{g}$, understanding their fatigue and creep response under these conditions becomes imperative. Not only does their behaviour change with the absolute value of temperature but also with its variation and cycles [31]. Hence, to better understand the effects of temperature on polymers, we should take a look at the molecular scale. It is known that when the temperature rises, there is an increase in the total energy of the matrix due to the vibration of molecules and chain segments. There are different types of molecular movements depending on the temperature and size of the structural units being considered [31]. At $T_{g}$, chains in amorphous regions (Figure 8) start to move relative to each other, occasioning an important decrease in stiffness (Figure 9). Crystalline regions, however, do not present the same behaviour and act as a reinforcement of the matrix through their structurally ordered lamellae (Figure 8). For semicrystalline polymers, there is a light softening and rubbery behaviour between $T_{g}$ and the melting temperature $T_{m}$, owing mainly to the amorphous phase present [32]. Only when the temperature reaches $T_{m}$ does the crystalline part melt and the material liquefies, behaving, thus, as a viscous liquid (Figure 9) [32].

We also note that the fatigue resistance of polymer-matrix composites generally increases with an increase in stiffness and/or strength [34]. Hence, considering how temperature affects the latter, it comes as no surprise that an increase in temperature also worsens fatigue performance. Additionally, due to the viscoelastic, i.e., *time-dependent*, properties of polymer-matrix composites, creep and creep–fatigue interactions are important factors that need to be accounted for whenever cyclic stresses under different temperatures are likely to occur [35,36].

As it was previously mentioned, a rise in temperature leads to more energy available for molecular chains to move, slide and uncoil. Since creep deformation is mainly characterised by these types of interactions, a rise in temperature is expected to lead to a more prominent creep effect (Figure 10) and to a reduced fatigue life [34].

On the one hand, thermoset matrices generally present a higher resistance to creep owning to the cross-linking of the polymeric chains. On the other hand, thermoplastics are more sensible to creep deformation; however, this behaviour can be improved by increasing the degree of crystallinity of the microstructure, which consequently hampers the sliding of chains [34,38]. Hai C. Tang et al. [39] have estimated a way of determining the slope *m* of an *S*-*N* curve as a function of the temperature level:(1)$$m=m_{0}exp\left(\frac{Q}{RT}\right),$$
where *Q* is the activation energy, *R* the gas constant and *T* the temperature in Kelvin.

The increase in temperature results in a lower number of life cycles (Figure 11). H. Mivehchi and A. Varvani-Farahan have also registered in their review [30] the same principle, not only regarding *S*-*N* curves but also for fatigue damage, having concluded that with higher temperatures the fatigue damage evolution increases.

Khan et al. [41] ran fatigue tests on plain weave woven carbon fibre–epoxy prepregs having not only concluded that there is a reduction of fatigue life with temperature (due to the augmented viscoelastic behaviour and probably weakened fibre–matrix bonding) but that this effect is less noticeable for lower stress amplitudes (high-cycle regions).

Relating to high-cycle fatigue, there is an essential factor to take into consideration, namely autogenous heating, which is prone to occur in polymers with higher non-linear behaviour and damping capacity, such as thermoplastics [42]. Their low-temperature conductivity [35,42] leads to a temperature increase, in turn, reducing the fatigue life (Figure 12).

Despite the fact that fatigue life is generally better with lower temperatures (mainly due to higher stiffness), when extreme levels are considered, some aspects need to be considered [43]. When the temperature drops below a certain level, compressive residual stresses develop due to different coefficients of thermal expansion of the matrix and the fibre. These residual stresses, which develop mainly on the fibre-matrix interface, may influence the mechanical properties of the composite [44,45]. In some occasions, the compressive stresses are high enough to induce plastic deformation and micro-cracking, the latter responsible for allowing the nucleation of macro-cracks (Figure 13) and the reduction of both stiffness and strength.

Thermoplastics, contrarily to thermosets, do not present any crosslinking, allowing for better relaxation of residual stresses [45]. However, multiple crack formation favours energy absorption and ameliorates the fatigue behaviour due to the splitting of fracture energies and stress concentrations [46]. Since failure mechanisms at cryogenic temperatures are mainly due to different thermal expansion coefficients, the layup and direction of the fibres are key factors for the cryogenic life of composites [47], being difficult to empirically predict a general behaviour of PMCs under these extreme conditions. On several applications, composites are subjected to high-temperature variations such as supersonic flight (between −55 and 120 ${}^{{}^{\circ}}$C) [48] or space vehicles/environments (between −150 and 150 ${}^{{}^{\circ}}$C) [49], which can lead to undesirable effects such as the appearance of cracks and degradation of mechanical properties [49,50,51]. Specifically, crack formation can occur either by differences in the temperature expansion coefficient of the fibre and that of the matrix [49] or between layers of the laminate, the latter case being the predominant one [51]. It has also been demonstrated that the degradation of the mechanical properties is greater in the first number of cycles, decaying asymptotically with an increasing number of cycles (Figure 14) [50,51]. In addition, the higher the order of thermal shocks, i.e., greater and/or faster temperature variations, the higher is the probability of these effects to happen, as well as their severity [52].

## 7. Lightning Strike

Lightning strike is a natural occurrence to which composite structures (wings, wind turbine blades, etc.) are exposed to in an open field environment. Due to the electrical properties of polymers, damage (which sometimes cannot be detected through visual inspection) is prone to occur whenever these components are struck. Therefore, evaluating the effect of this occurrence on the structural integrity and fatigue life of polymer-matrix composite parts is essential [53,54]. Whenever a composite structure is struck by lighting, a very high current channel is created and absorbed by the material [53]. Polymeric matrices usually present poor conductivity, leading to a large temperature increase in zones neighbouring the attachment point due to the Joule effect [54]. The achieved temperatures go above the pyrolisis gasification temperatures of the polymer matrices, occasioning the formation of gases. Due the heat ablation, the accumulation of these gases inside small interlayer delamination spaces originates an explosive effect, further intensifying delamination as well as shock-wave effects [54,55]. In Figure 15, a representation of the lightning-induced delamination is presented.

With the high level of delamination, crack growth mechanisms and matrix degradation, the fatigue life of composites subject to these types of events is drastically reduced. Despite the majority of the damage being caused by these factors, many other effects need to be considered, such as fibre breakage, quick changes of electric and mechanical properties as well as very high-temperature gradients through different directions and layers [54,55], making this type of event challenging to simulate using commercial finite element software [53,55]. Despite some solutions that have been proposed to mitigate the problem, such as the inclusion of electrical conductive nanoparticles in the polymer or the use of better conductive matrices, their maturity and ease of manufacturing are still the limiting factors that hamper their wide-scale implementation [56].
**Part II—Polymer types**

## 8. Thermoset Polymers

Thermoset polymers are composed of chemically covalent bonds—cross-linked molecules in a network structure. On heating, they decompose—as opposed to softening—and do not undergo a liquid state. Once solidified by the cross-linking process, they cannot be reshaped. However, cross-linking results in great strength and rigidity owing to the restriction of the relative displacements of the macromolecules. Among commonly used thermoset matrices with continuous fibres, we identify epoxy, vinyl ester and polyester resins [57,58,59], which are the focus of this section. A summary is presented in Table A1 in Appendix A, wherein the studies mentioned throughout this section are compiled.

### 8.1. Epoxy

Epoxy resins are one of the major thermoset matrix materials, featuring two or more oxirane rings or epoxy groups as part of its chemical structure [57,60]. Fibre-reinforced epoxy (FRE) composites are extensively used in a broad spectrum of industrial applications such as aerospace, marine, and automotive, where durability, reliability and performance are required under hostile environments [61]. Hence, understanding its behaviour concerning fatigue and creep under various environmental conditions is pivotal in designing safe and dependable structures.

#### 8.1.1. Water

The performance of FRE composites under moisture exposure is crucial, and it has been widely established that moisture has significant repercussions on the physical and chemical properties of such materials [61,62,63,64,65], owing to a large number of highly polar specific functional groups in the cured epoxy resin resulting in hydrophilic materials, i.e., with high water affinity [63]. Water uptake is correlated to mechanical performance losses mainly due to plasticising; however, hydrolysis reactions and water acting as a physical cross-link may occur as well [64,66]—refer to Section 2. The ramifications of such mechanisms are not confined to the static material properties and have been recorded to detrimentally affect the fatigue performance of epoxy-based composites. Tensile fatigue lives of glass/epoxy specimens were found to be reduced from $10^{7}$ to $10^{5}$ cycles when loaded at 65% of their ultimate tensile strength (UTS) immersed in distilled water at 25 ${}^{{}^{\circ}}$C [67]. Hybrid glass-carbon/epoxy samples showed similar fatigue performances in wet and dry conditions of up to 45% UTS before a noticeable drop-off was registered [67]. Ageing of composite materials in aqueous environments also has prejudicial consequences on the mechanical performance: unidirectional carbon/epoxy laminates have been found to withstand more than 3 × 10${}^{6}$ cycles at 80% of the ultimate flexural strength (UFS) when subjected to bending fatigue tests without preconditioning. Nevertheless, aged wet laminates—immersed in freshwater, seawater and seawater with 70 bar hydrostatic pressure at 50 ${}^{{}^{\circ}}$C for 3 months—could not achieve comparable lifetimes even at 65% UFS [68]. Although the maximum creep strain of glass/epoxy specimens was found to actually increase for immersions times under 24 h due to a reactivation of curing reactions, the effects of water are evident in a gradual reduction up to a minimum value corresponding to 504 h of immersion as per Figure 16.

#### 8.1.2. Temperature

It is widely established that the mechanical properties and structural integrity of epoxy-based composites are significantly reduced [70,71] due to an immediate softening and rubber-like response at elevated temperatures—even only moderately above $T_{g}$ [61]—restricting their use in high-temperature applications [34,72]. At temperatures considerably exceeding $T_{g}$, epoxy resins in general begin to decompose. The FRE composites initially undergo reversible mechanical changes with rising temperatures close to $T_{g}$, such as viscous softening [61]. As a consequence, properties of both the polymeric resin and fibre–matrix interface can be adversely affected which is accompanied by a significant loss of mechanical performance [61]. This is evidenced in the fatigue life of 3D angle-interlock woven carbon/epoxy composites tested at 25 and 100 ${}^{{}^{\circ}}$C, where a drop from 38% UTS to 30% UTS in the fatigue limit was observed with increasing temperature [73]—attributed to weakening in the interface bonding strength. Basalt/epoxy specimens were also recorded to have significant performance losses with increasing temperature concerning both fatigue and tensile properties—a linear decrease from −20 to 60 ${}^{{}^{\circ}}$C with a reduction of 20% in UTS [74]. On the other hand, an increase in mechanical properties with decreasing temperature was also recorded. Mitsuhiro Okayasu and Yuki Tsuchiya [75] also reported an enhancement of the low-cycle fatigue performance by a factor of 1.5 on a unidirectional carbon/epoxy specimen at cryogenic temperatures. This factor decreases with the increasing number of cycles and at the later phase, the fatigue curve at −196 ${}^{{}^{\circ}}$C is close to the one at room temperature. This could be attributed to a magnifying effect at low temperatures on chain movement inhibition due to high cross-link density as previously reported [76,77] and as stated by the authors.

#### 8.1.3. Hygrothermal

Hygrothermal environments stem from the combined effects of water and temperature. In addition to the water uptake, a rise in temperature is capable of significantly accelerating the saturation of FRE due to a superior rate of diffusion. The enhanced chain mobility also exacerbates the drop in mechanical performance [61,78]. Irreversible chemical changes may also be induced in the material, i.e., that persist even after the composite is dried: a permanent performance loss of up to 17% has been reported on preconditioned glass/epoxy composites used in wind turbine blades—immersion in demineralised water at 50 ${}^{{}^{\circ}}$C for 4800 h [79]. The combination of moisture and temperature also resulted in shorter fatigue lives by up to three orders of magnitude. The drop is fatigue performance is also evident on the fatigue tests of a wet epoxy resin—distilled water at 60 ${}^{{}^{\circ}}$C for 2 months—for which the *S*-*N* curve was shifted by 20% whilst maintaining the same slope [80]. Hygrothermal ageing of glass/epoxy specimens in ambient, humid (40 ${}^{{}^{\circ}}$C and 60% relative humidity, RH) and sub-zero specimens (−15 ${}^{{}^{\circ}}$C) for 180 days resulted in a decrease from $10^{6}$ cycles at 40% UTS to 5.6 × 10${}^{5}$, 5.8 × 10${}^{5}$ and $3.4\times 10^{5}$ for each respective conditioning process [81]. Significant deterioration in fatigue behaviour was attributed to leaching of the glass fibres and hydrolysis of the epoxy resin by E. Vauthier et al. [82] on specimens aged in various temperature and humidity conditions. In addition, findings on the fatigue response of preconditioned natural FRE composites yielded similar outcomes. Immersion in distilled water at 70 ${}^{{}^{\circ}}$C for 26 days was reported to induce noteworthy drops in the fatigue response of epoxy resins reinforced with flax fibres for the low-cycle range—for 20,000 cycles, the maximum stress is reduced from 196 to 148 MPa [83]. However, the obtained slopes of the fitted curves—9.4 for aged samples vs. 18.9 for unaged ones—point to an enhanced fatigue resistance for high numbers of cycles, seeing as both curves converge to 90 MPa for a lifetime of $5.3\times 10^{6}$ cycles, and it has been proven that, for the material in question, the trend of the *S*-*N* curves is the same up to $10^{8}$ cycles [83,84]. On non-woven flax/epoxy specimens severe degradation was also identified—at 50% loading level, the fatigue modulus loss increased to 70% for preconditioned samples in a water bath at 45 ${}^{{}^{\circ}}$C from 25–30% for unaged ones [85].

#### 8.1.4. Seawater

Through plasticisation and hydrolysis, seawater exposure in FRE composites can severely damage the thermophysical, chemical, and mechanical performance of the epoxy matrix [61,86]. Fatigue life predictions revealed that, for $10^{7}$ cycles, basalt/epoxy composites aged in seawater for 6 weeks would undergo a reduction of 11% concerning the maximum applied stress [87]. Notched carbon/epoxy laminates—rectangular cross-sections with a central hole—suffered dire repercussions subsequent to immersion in natural and artificial seawater for 30 and 60 days in their low-cycle fatigue behaviour—for $10^{4}$ cycles, the stress amplitude was 1.2 times higher for dry laminates [88]. Still, no discernible differences were observed in the fatigue behaviour regarding the immersion duration or the type of seawater and for high-cycle fatigue, the lifetime of the laminates was not affected—superior impact of holes and respective induced stress concentrations.

#### 8.1.5. Liquid Chemical Agents and UV Radiation

FRE composites are often employed in outdoor applications where the material is subjected to corrosive medium and ultraviolet (UV) radiation in addition to moisture and temperature [61]. Concerning the former, alkaline solutions have been shown to severely diminish the fatigue life of pultruded carbon/epoxy specimens following a prolonged immersion for a 2-year period. Fatigue life predictions resulting from several methods, namely, *S*-*N* curves, Whitney’s method and Sendeckyj’s method, identified a reduction in the maximum allowable stresses for 2 million cycles from 69.28 to 60.75% UTS, 73.39 to 68.55% UTS and 65.00 to 54.41% UTS for each respective method [89]. E-glass/epoxy specimens aged in aqueous solutions of NAOH having concentrations of 5%, 15% and 25% for 1, 2, 3 and 4 months suffered a reduction of 21.36 and 62.38% in fatigue strength and life [90]. The same procedure applied to an HCl aqueous solution resulted in more severe losses of 41.95% and 97.19% [90]. On the other hand, the energy associated with the wavelengths—290 to 400 nm—of the solar radiation incident on the surface of the earth is comparable to the breakdown energies of covalent bonds of epoxy and consequently can result in discolourations, chain scission-induced microscopic degradation and reduction in mechanical properties [61,91]. The exposure of tri-axial carbon/epoxy laminates 313-bulb UV for 750 h had major consequences on fatigue life for stress ranges smaller than 500.0 MPa with a major decrease from above $10^{5}$ cycles to below $10^{4}$ cycles [92]. Exposure to γ-rays corresponding to doses of 20, 100 and 200 kGy also detrimentally impacted the fatigue performance of glass/epoxy specimens, particularly at low-cycle fatigue ($N<10^{4}$) where a gradual decrease in the maximum stress was identified with increasing irradiation doses. This also translated into fewer life cycles for a given stress value with increasing doses. Nonetheless, no drops in the fatigue life were detected at high-cycle fatigue stages ($N>10^{4}$) [93].

### 8.2. Vinyl Ester

Vinyl ester resins originate from an addition reaction between and epoxy resin and an unsaturated carboxylic acid such as acrylic or methacrylic and boast great mechanical performance comparable to epoxy whilst simultaneously offering superb processability similar to a polyester resin [60].

#### 8.2.1. Hygrothermal

The combined effect of moisture uptake and elevated temperatures has already been established to degrade the fatigue life of fibre-reinforced epoxy polymers. This is extended to vinyl ester resins, with immersion in water at 40 ${}^{{}^{\circ}}$C for periods ranging from 30 to 600 days being proven to significantly deteriorate the fatigue performance of glass/vinyl ester laminates [94]. Still, while the laminates showed a greater loss in fatigue life at the superior stress levels—70%—the vinyl ester resin samples showed more degradation at 30% of the ultimate stress. The findings revealed that for brief exposure intervals, the fatigue behaviour for composite materials is independent of the moisture content. However, the impact becomes more apparent with prolonged exposure.

#### 8.2.2. Seawater

Exposure to aqueous environments has been shown to severely impact the fatigue performance of glass/vinyl ester composite pipes [95]. The fatigue strength of corroded specimens due to salt water spray was found to be decreased from a stress amplitude of 24 MPa for $10^{6}$ cycles to 15 MPa after exposure for 300 h. Immersion in oxygenated seawater also displayed detrimental consequences to fatigue performance with the effect being more prevalent with increasing exposure periods. This deterioration effect was particularly obvious in the low cycle zone, where specimens exposed for 10,000 h saw a reduction in fatigue life by a factor of 6. Exposure to 100% RU of up to 3000 and 10,000 h elicited a slight performance loss, with the repercussions being exacerbated in the low cycle region where the fatigue strength and life were respectively reduced by ∼2 MPa and a factor of 2. The impact of exposure to both fresh and saltwater on the fatigue durability of the glass fibre-reinforced/vinyl ester composite was also characterised by McBagonluri et al. [96]. Both environmental circumstances have been observed to result in a significant decrease in fatigue strength. The authors come to the conclusion that water intrusion in both freshwater and saltwater conditions causes the load-bearing fibres to gradually deteriorate, which leads to fatigue failure. The flexural fatigue life of aged—exposure to water and salt solutions containing mass fractions of either 5% NaCl or 10% NaCl for up to 6570 h—pultruded glass/vinyl ester composites was recorded to be severely reduced from beyond $10^{7}$ cycles to within $10^{6}$ to $10^{7}$ cycles at 30% of the flexural strength of the specimens [97].

Similar results were reported for carbon fibre-reinforced vinyl ester resins, with fatigue lives of fully saturated woven carbon/vinyl ester—immersed in synthetic seawater for a period of 140 days—evidenced a decrease up to ∼62% across all strain ranges with this effect being more prevalent for higher strain ranges—low-cycle fatigue—where the average measured reduction is between 37% at 0.46% strain range and 90% at 0.6% strain range [98]. Fatigue lives of carbon/vinyl ester laminates tested immersed in seawater induced a staggering 71% degradation in the number of cycles to failure when compared to dry specimens, whereas simply preconditioning—immersion in seawater at 40 ${}^{{}^{\circ}}$C for 6 months—resulted in a drop of 30% [99].

#### 8.2.3. Liquid Chemical Agents

Under liquid chemically active environments, PMCs are expected to corrode and to suffer losses in their fatigue response. This is evident in the striking 58% decrease in the slope of the *S*-*N* curves for unaged basalt/vinyl ester tendons sprayed with artificial acid rain—pH of 2.5. This was attributed to the strong corrosive properties of the acid that, with penetration, induced chemical corrosion in the constituents of the PMC in addition to the internal stresses originating from fluid ingress [100].

### 8.3. Polyester

Polyester macromolecules are attained from the reaction between a diacid or dianhydride with a dihydroxy compound. Numerous types of sail and motor boats, fishing boats, and naval vessels extensively employ polyester resin-based fibre-reinforced composites in the construction of their hulls [60]. Hence, understanding their environmental fatigue in such an environment is of the utmost importance.

#### 8.3.1. Water

Apart from seawater, moisture uptake has already been established to detrimentally impact the fatigue life of fibre-reinforced epoxy polymers. This is extended to polyester resins, wherein tensile fatigue lives of polyester-based GFRP laminates were recorded to decrease to one-ninth of the dry samples’ ones for a stress level of 40% UTS following a moisture uptake of 0.38% with respect to the initial weight of the material [101].

#### 8.3.2. Seawater

The effects of seawater on polyester-reinforced polymer composites have also been well established [102,103]. Reductions in fatigue performance were recorded and established to be exacerbated with increasing immersion times for randomly oriented short glass fibre-reinforced polyester composites immersed in natural seawater for 90, 180 and 270 days [102]. Furthermore, fatigue lifetimes under a simulated marine environment for preconditioned—immersion for 30 days in a 3.5 wt% NaOH solution—non-crimp wound glass fibre-reinforced polyester composites were recorded to be significantly shortened. Still, the same failure mechanisms were detected for both dry specimens tested in air and preconditioned ones tested in artificial seawater—the *S*-*N* curves exhibited the same slope [103].

## 9. Thermoplastic Polymers

Thermoplastic matrix composites have long been used for several different varied and ordinary applications; however, there has been an increase in investment and development of technologies for their use in more demanding scenarios, such as the aerospace industry. They boast higher matrix toughness and impact resistance, faster and more flexible production cycles as well as a better recyclability, crucial considering a more sustainable industry. However, their needs for high-temperature processing, poor fibre impregnation and high sensibility to heat come as disadvantages, deeming their implementation difficult [104,105,106]. Contrarily to thermosets, thermoplastic polymers may present a crystalline phase, whose degree may vary accordingly to the cooling rate from the melting temperature [34]. Biron [105] divided a wide range of thermoplastics into three different types of usage: commodities, technical thermoplastics and speciality thermoplastics. In this section, a thermoplastic matrix from each type (PP, PA and PEEK) is presented and, for each polymer, its resistance to environmental factors reviewed from the available data in the literature. As for the case of thermoset matrices, the studies mentioned in this section are gathered in Table A2 (Appendix A).

### 9.1. Polypropylene

Polypropylene is a partially crystalline thermoplastic used for a wide range of applications (automotive, appliances, furniture, etc.) [31], featuring relatively good mechanical properties at ambient temperature as well as good dimensional stability, flame resistance and fatigue resistance [31,107]. Furthermore, PP has been appointed as a very promising matrix for the use of natural reinforcements [107].

#### 9.1.1. Water

As previously mentioned, water absorption by polymer-matrix composites leads to mechanical and chemical changes that pose a problem to the life of several components. Polypropylene matrix composites have been documented to have their fatigue life reduced with an increase in moisture content [108,109]. Ayman et al. [109] conducted tests on wet short glass fibre-reinforced PP samples and concluded that there is a reduction of fatigue life when exposed to moisture (Figure 17).

This phenomenon is once again explained by the influence of water on the interfacial strength between fibre and matrix. The effect is reduced by increasing the reinforcement content; nevertheless, the fibres are also subject to corrosion, which leads to premature fracture with increasing water content [109].

#### 9.1.2. Temperature

Similarly to other PMCs, PP matrix composites face a reduction of fatigue and creep life with an increase in temperature [110,111]. Viña et al. performed several fatigue tests, at different temperatures, on sheets of glass fibre-reinforced polypropylene.

It can also be observed in Figure 18, where the strong viscoelastic behaviour characteristic of thermoplastics leads to high non-linearity of the *S*-*N* curve at high temperatures [111].

#### 9.1.3. Hygrothermal

Moisture absorption and temperature are related to each other more often than not. When exposed to higher temperatures, water diffusion is easier and enhances the moisture degradation capability. In Figure 19, the influence of temperature on the moisture absorption of a hemp-reinforced PP can be observed [110,111]. For a temperature level of 80 ${}^{{}^{\circ}}$C, the moisture uptake can be up to 3600% faster than when subject to ambient temperature. Naturally, the moisture uptake rates are highly dependent on the fibre content and nature [110,112,113], especially for natural fibres (hydrophilic by nature) [114].

#### 9.1.4. Seawater

Polypropylene matrix composites are being considered for use in civil construction and others under marine environments. Being so, it is necessary to assess how the salinity affects the behaviour of such materials [116,117]. Dong et al. [116] have tested sheets of glass fibre-reinforced polypropylene underwater with different saline concentrations for a duration of 6 months. It was concluded that, under seawater, the water uptake increases for the same interval of time compared to distilled water (Figure 20). The corrosion of glass fibres was also evident, with fatigue and creep resistance naturally degraded by the influence of these mechanisms [116].

Even though the same effect was registered by other authors such as Najafi et al. [117] with wood–polypropylene samples, there is still limited research regarding the behaviour of polypropylene composites under saltwater environments, mainly due to the high dependence of the response on different fibre types and contents.

#### 9.1.5. UV Radiation

The effect of UV radiation on the fatigue life of different polypropylene composites has been documented by Joseph et al. [118] in their research. An important reduction of the mechanical properties of PP composites when exposed to UV radiation for long periods was observed, which occurs mainly due to chain scission (photo-oxidation) and to the appearance of cracks due to chemi-crystallisation (Figure 21).

Nasri et al. [119] have also studied the influence of UV radiation on the mechanical properties of PP composites. Using specimens of flax or pine fibre-reinforced PP, the response to a flexural test after different exposure times was carried out. It was concluded that the longer the exposure time, the more the mechanical properties are degraded, both in stiffness and strength.

### 9.2. Polyamide

PA is a thermoplastic used as a matrix mainly for short-fibre-reinforced composites in a wide range of applications. The low density, good mechanical properties, heat stability and the wide range of processing techniques, including additive manufacture [120], make this material highly desirable for a vast number of engineering solutions [31,121]. Among the different polyamides available, the most used are PA6 and PA6,6 belonging to the aliphatic polyamide family, also known as nylons [31,122].

#### 9.2.1. Water

It is widely established that polyamides present a significant hygroscopic behaviour [31,123]; therefore, some research has been conducted in order to evaluate the mechanical response of PA composites under aqueous environments. Barbouchi et al. [124] carried fatigue tests on short glass fibre-reinforced PA6,6 samples on atmospheres with different % RH. Besides registering a steep decrease in the glass transition temperature $T_{g}$, there was an important reduction of both stiffness and strength. Consequently, the fatigue behaviour was diminished by the water intake; however, the fatigue endurance limit was observed to be the same for both dry and wet samples. The same substantial reduction of mechanical properties was observed by several other authors [125,126,127]. Water uptake, up to 8.5 wt%, decreasing with the amount of fibre content, was found to have a significant effect on the fatigue of the PA composite samples. These changes are explained by the plasticising effect of water which acts mainly on the amorphous phase, predominant in the PA microstructure [126].

#### 9.2.2. Temperature

Due to the high presence of an amorphous phase in some PA composites, mainly due to the high cooling rate verified during manufacturing, the mechanical response is expected to be highly affected by an increase in temperature. Kawai et al. [128] evaluated the fatigue response of short carbon fibre-reinforced PA6 samples at different temperatures (room temperature, 50 ${}^{{}^{\circ}}$C, 70 ${}^{{}^{\circ}}$C). It was concluded that the fatigue strength for the same number of cycles decreases with an increase in temperature. The results are on par with the substantial stiffness and strength reduction (approx. 50%) that were verified under higher temperatures. Monte et al. [129] documented a similar level of reduction in strength and fatigue resistance with higher temperatures, adding that the characterisation of tensile-compressive fatigue was very difficult due to instability problems, since at *T* = 130 ${}^{{}^{\circ}}$C, the samples were immensely compliant. Hysteric heating was also verified by previous research to further decrease the fatigue performance of PA composites [129,130].

#### 9.2.3. Hygrothermal

In the previous section, both temperature and water uptake were shown to be detrimental to the properties of PA matrix composites. In order to establish a relation between the two variables, Sang et al. [131] ran different tests to evaluate the moisture uptake with the temperature. Moisture uptake increased in a Fickian way reaching the plateau at the maximum moisture uptake. The same results were obtained by Lei et al. [132], who also documented an important degree of swelling and reduction of mechanical properties. (Figure 22).

The use of PA matrix composites in hot and humid environments is then discouraged since the high content of the amorphous phase is heavily influenced by environmental conditions.

#### 9.2.4. Seawater

In order to evaluate the effect of salt concentration in the moisture uptake or degradation of PA matrix composites, Haadar et al. [133] immersed short glass fibre-reinforced PA in different environments, distilled water and saline solution. The water uptake was observed to evolve in a similar way for both conditions until near the saturation level, where the presence of sodium chloride slightly decreased the water diffusivity. The impact on the mechanical properties was similar, with NaCl-immersed specimens retaining a slightly higher percentage of their dry properties than the ones immersed in distilled water. Therefore, despite inducing a small change, fatigue and creep behaviour between distilled and saltwater environments are not significantly different.

#### 9.2.5. UV Radiation

The assessment of the response of polyamide composites to UV radiation is a necessity for natural light-exposed engineering applications. Markovičová et al. [134] conducted some experiments to evaluate the UV radiation effect on glass fibre-reinforced PA, concluding that, up to 750 h of exposure, there was a moderate loss of tensile strength and flexural stiffness; nonetheless, there was recovery after 1000 h, which was attributed to partial cross-linking of the matrix. Furthermore, Pinpathomrat et al. [135] observed that unreinforced PA6 had a significant loss of approximately 85% of tensile strength after being exposed for 7 days, yet the same behaviour was not obtained for the reinforced counterpart [136]. A clear influence of UV exposure on the fatigue and creep behaviour of PA matrix composites requires more research.

### 9.3. Polyetheretherketone

PEEK is a partially crystalline polymer belonging to the polyaryletherketone (PAEK) family, constituted by high-performance thermoplastics [34,137]. Its stability at high temperatures (300 ${}^{{}^{\circ}}$C), compatibility with high-strength fibres, high strength, stiffness and good resistance to chemicals and radiation make it especially attractive for more demanding applications such as the aerospace and medical industries [31,34,137].

#### 9.3.1. Water

Determining the behaviour of PEEK polymers with different water contents becomes a necessity when a designed component is made to last in any environment. After three weeks of immersion, a weight change on the uptake equilibrium on boiling water of 0.45% was recorded [138]. At 65% RH, equilibrium was attained only after 1000 h, with the specimen exhibiting a weight change of only 0.08%. After performing the fatigue test analysis, it was concluded that the effect of the moisture absorption had no relevant effects on the fatigue performance of the composite (expected behaviours due to the low moisture absorbance). This was further verified by other studies [139], which also appointed a low moisture uptake as the main reason behind the maintenance of the fatigue performance.

#### 9.3.2. Temperature

The use of PEEK polymer-matrix composites for high performance requires the assessment of its behaviour under different temperatures. Studies have reported that an increase in temperature leads to a degradation of both longitudinal stiffness and strength (especially near the glass transition temperature) of the samples, mainly attributed to the movement of chains in the amorphous parts [140,141]. Consequently, there is a reduction of the fatigue life of the composite (Figure 23).

The same behaviour was also documented by Kawai et al. [141], particularly on the effect of temperature on the off-axis loading of unidirectional AS4/PEEK samples. The failure surfaces showed tearing of the matrix, probably due to the higher shear stresses acting along the off-axis direction. The effect of temperature on creep was also studied [142], and a rise in the tensile creep compliance with higher levels of temperature was registered, showing excellent creep resistance at room temperature everywhere but near the glass transition temperature (above 130 ${}^{{}^{\circ}}$C), where creep compliance rises sharply [143,144]. Both the fatigue and the creep-temperature responses of PEEK composites are highly dependent on the matrix, mainly due to its viscoelastic behaviour at high temperatures [142,143,144].

#### 9.3.3. Hygrothermal

Exposure to higher temperatures leads to a higher moisture intake, with the mechanical properties such as flexural strength experiencing a slight reduction, up to 15%, under 80 ${}^{{}^{\circ}}$C at 85% RH [145]. Several other studies [138,146,147] concluded that the mechanical properties of PEEK composites subject to different temperatures and moisture contents do not degrade in a substantial way, not being, however, totally negligible.

#### 9.3.4. Seawater

Boinard et al. [148] exposed samples of PEEK to different combinations of temperature and salt content (distilled water and brine) and concluded that neither the absorption nor the diffusion process are affected by the presence of sodium chloride. Schambron et al. [149] obtained the same results in their research on carbon-fibre/PEEK bone plates. Mechanical and fatigue tests after exposure to different ageing environments with several NaCl concentration levels also showed that the degradation of mechanical properties is hardly affected by immersion in water or saline environments at room temperature and that the fatigue life is not altered.

#### 9.3.5. UV Radiation

PEEK matrices have shown little to no reaction, in mechanical properties, to UV exposure. Nakamura et al. [150] evaluated the mechanical properties of PEEK sheets after being exposed to UV radiation, and recorded no significant reduction in tensile strength. A slight increase resulting from the embrittlement has been registered due to the UV-induced crosslinking [151]. This increase in strength is inversely proportional to the static tensile stress applied to the specimens while being exposed. Niu et al. [152] registered a decrease of 7.5% of stiffness after 1560 h of alternating cycles between 8 h of UV irradiation of 1.55 ${W/m}^{2}$ at 70 ${}^{{}^{\circ}}$C and 4 h of dark condensation at 50 ${}^{{}^{\circ}}$C, having identified fibre/matrix debonding near the surface. This reduction in stiffness may slightly compromise the fatigue life of PEEK composites, especially with higher degrees of crystallinity. Regardless, for a significant reduction of mechanical properties to happen, very demanding and specific conditions are needed, showing that PEEK has good resistance to UV radiation for the majority of applications, much due to its stable aromatic structure, which makes it completely inert to a wide range of chemical environments [153,154].

## 10. Biopolymers

As we thrive to reduce today’s society’s environmental footprint, adopting more and more sustainable practices, the polymer industry has concomitantly reacted with the introduction of the so-called *biopolymers*, also referred to as *biodegradable* or *bio-based*/biogenic [155]. Despite being interchangeably used, they entail different meanings: *bio-based*/biogenic points to the origin of the polymer or the raw material from which it was produced, specifying materials, generally, macromolecular polymers, that derive exclusively from renewable sources, whereas *biodegradable* depends only on the chemical structure of a polymer and refers to the recycling options available at the end of its service life, meaning they can be fully degraded by microorganisms or enzymes under ambient conditions (i.e., outside composting plants) within a reasonable time length resulting in nontoxic products. These two terms are, thus, independent of each other and should not be confused, e.g., polymers based on fossil resources can still be biodegradable, and bio-based polymers are not necessarily biodegradable [25,155].

In the following sections, we focus on some of the most representative bio-based and biodegradable polymer matrices, namely starch—and its derivatives, particularly TPS, PLA and PHAs—the most commonly used being PHB. On one hand, starch and PLA are, by far, the most productive biodegradable plastics, accounting for 38.4% and 25.0% of the total biodegradable plastic capacity, according to European Bioplastics Data from 2019 [11,156], with PLA totalling 449,000 tons per year, its applications ranging from packaging to medical devices. On the other hand, PHAs, despite their small market, high production costs and complex extraction processes, are regarded as the most promising biodegradable polymer—with a market worth of USD 57 million in 2019 almost doubling to USD 98 million in 2021 (projected), being not only biodegradable and naturally produced, but also biocompatible with many organisms [11].

Additionally, to the best of the authors’ knowledge, there are very few works focusing on the environmental fatigue and creep behaviours of these polymers—let alone their composites—which comes as no surprise, considering that the shift of focus towards these materials happened only recently, and the experimental methods are yet to mature. As we detail below, most of the literature verses *only* on their static degradation, and, even if that deviates from the scope of this review, it is the authors’ belief that there is still insight to be gained from their degradation behaviour. Faster degradation kinetics (in strength and moduli) have been linked with increased deformation—*strain*—levels [157], and, since the mechanisms actuating in environmental fatigue and creep are the same as those involved in degradation, mechanical tensile/compressive loading is only expected to accelerate/delay the process. Finally, Table A3 (Appendix A) gathers the studies mentioned throughout this section, briefly presenting their key findings.

### 10.1. Polylactic Acid

PLA is a polyester obtained from agroresources that is chemically synthesised using monomers from the fermentation of plant-derived carbohydrates [13,25,158], which is nearly amorphous in its pure form. Due to its low glass transition temperature (50–80 ${}^{{}^{\circ}}$C), in several applications, it is often reinforced with fibres—often organic, such as kenaf or sisal—or particles. As mentioned, very few studies have been performed regarding PLA mechanical, let alone environmental, fatigue behaviour [159,160,161,162], most of which focus on the properties of additive manufacturing specimens [163,164,165,166,167,168,169,170,171,172,173] and some in PLA composites [174,175] and creep behaviour [162,176].

#### 10.1.1. Water

PLA boasts a low water and oxygen acceptability level [177,178], with degradation occurring due to hydrolysis of ester linkages followed by microbial breakdown [179,180]. This preferentially affects the amorphous regions of the polymer by erosion—only after a *long time* do the crystalline regions undergo hydrolysis—and the degree of crystallinity of PLA samples has been found to increase with degradation time [181]. Additionally, the pH value was coupled with the degradation rate, and the degradation products themselves contributed to altering the pH of the medium; more specifically, he decomposition rates increased with increasing pH in the range 3.4–7, yet all samples assessed, both semicrystalline and amorphous, attained the same pH value after about 20 days of hydrolysis [181].

*D*-PLA, despite being hydrolysable, is considered non-degradable, contrarily to the degradable *L*-PLA [25,182,183,184]. Several studies have reported a negligible (or even no) mass loss when immersed in sterile water (Figure 24) [13,185], even immersed in a laboratory aquatic medium over a period of 112 days (constant light, 30 ${}^{{}^{\circ}}$C) [25].

Regarding composites, moisture absorption has been correlated with the volume of fibres and voids in PLA/hemp composites [5], and although in the neat polymer a certain time is required for the water to reach the inner core, composite samples are quickly filled with water, leading to homogeneous hydrolysis [186]. Likewise, PLA/TiO${}_{2}$ nanocomposites were tested at (37 ± 1) ${}^{{}^{\circ}}$C and pH = 7.4 over more than one year, and it was found that the TiO${}_{2}$ particles accelerated the degradation kinetics, with the hydrolysis starting at the interface between fillers and matrix [187]. It has been found for PLA/PEG gels that the loading frequency does not affect the degradation rate for blends with a low cross-link density, whereas a higher frequency is linked to faster degradation rates in highly cross-linked polymers [188].

#### 10.1.2. Temperature

PLA has a glass transition temperature close to 60 ${}^{{}^{\circ}}$C, below which the migration of moisture and microbial agents into the material is hampered by the increased stiffness and reduced mobility of the chains, slowing down its degradation rate [5]. Contrarily, temperatures above 50 ${}^{{}^{\circ}}$C have been found to accelerate the abiotic hydrolysis of PLA [189,190,191,192], even if its biodegradation kinetics are generally considered slow, [193]—for a more comprehensive review, see Karamanlioglu et al. [194]. Temperature has been determined to be the main factor affecting biodegradation in microorganism-rich environments such as compost, soil and seawater with faster degradation rates registered therein than in their sterile counterparts [191], also accounting for the differences between the distinct environments—soil and aqueous media [25,190,192,195]. Samples tested at temperatures from 25 to 50 ${}^{{}^{\circ}}$C exhibited degradation effects ranging from no damage after one year to a mass loss of more than 40% just after 5 weeks [191,192], or, similarly, a decrease of 55% in tensile strength over 11 months [196,197].

Focusing on composites, hydrophilic fillers, e.g., starch, kenaf fibres or wood–flour, have been reported to increase the thermal decomposition rate under composting conditions for as long as 90 days, e.g., from 60% to 80% when the starch content increased from 10% to 40%, the same occurring when the $T_{g}$ is decreased by blending with other polymers [198,199,200]. In addition, in PLA/KBF and PLA/flax biocomposites, a larger loading was correlated with an accelerated decomposition, due to the increased micropore surface area which, in turn, increases the transporting channels inside the composite used by humidity and microorganisms [199].

#### 10.1.3. Seawater

PLA has been found to degrade very slowly in seawater, due to the lack of effective microorganisms that only allows for a (slow) abiotic hydrolysis to occur [13]. The high salinity in seawater compared to distilled water delays the diffusion process and the degradation rate [11,201]. Even after 1 year, no visible degradation was reported for the most common type of PLA (Figure 24) [11,185,201], and a weight loss of just 2.5% over 600 days was registered in a simulated marine environment [25,202].

Natural and static laboratory conditions have been compared, and the differences have been found negligible after a period of 10 weeks at 25 ${}^{{}^{\circ}}$C for various degrees of crystallinity (apart from the higher weight loss associated with mechanical forces) with a slight increase in stiffness and strength during the first weeks [203,204]. Specimens tested in a French harbour for 6 months exhibited only a reduction below 10% in tensile strength with laboratory tests concluding that seawater temperature significantly affects the mechanical properties [205].

However, even at moderate temperatures (around 40 ${}^{{}^{\circ}}$C), the tensile stiffness of PLA matrices in PLA/flax biocomposites has remained constant, with the stiffness and strength of the composite diminishing [206], behaviour attributed to wet ageing of the interfacial adhesion. For most organic reinforcements, the water uptake increases with the fibre content [207], leading, after long exposure periods, to the appearance of cracks in the fibre and in the matrix [206,208].

Finally, samples immersed in a phosphate-buffered saline solution have been tested and are shown to keep a strength above 200 MPa, from an initial bending strength of 280 MPa, for up to 25 weeks [186,209].

#### 10.1.4. UV Radiation

UV radiation has been shown to reduce the toughness of PLA/PHB samples due to surface modification, as it promotes solvent-induced crystallisation [177]. In a different study, samples exposed to natural sunlight for 90 days, despite exhibiting no discernible cracks or defects, suffered a 15–25% decrease in molecular mass and an increase in storage modulus [5,210]. Mass losses, changes in crystalline structures and degradation of thermal and mechanical properties, particularly Young’s modulus, were also noticeable under both laboratory and natural UV irradiation [210].

#### 10.1.5. Microorganisms

Microorganisms act in different environments, and throughout this section, we focus on natural soil and compost biotic media. Regarding the former, although PLA is considered naturally biodegradable, it hardly degrades in a natural environment, due to a lack of microorganisms capable of decomposing it [11,25,195,211,212,213], e.g., a weight loss of 5% after 180 days with no visible changes has been recorded by [25,214]. In compost, however, it takes about 6–9 months for its complete decomposition, especially since the temperature therein sits close to its $T_{g}$ (58–65 vs. 60 ${}^{{}^{\circ}}$C, respectively), promoting hydrolysis [11,157,191,192]. At 65 ${}^{{}^{\circ}}$C, samples have been recorded to suffer a mass loss of 34% after just 7 weeks [25,180], with further studies confirming the trend [13,191,215,216,217]. Moreover, the temperature has been shown to be the key factor affecting the polymer and the microbial communities present; for instance, in [218,219], the recorded degradation increased from 43% to 75% over roughly the same period when the temperature was increased from 37 ${}^{{}^{\circ}}$C to 55 ${}^{{}^{\circ}}$C.

Degradation happens in two steps (Figure 25): (1) hydrolysis—first chemical, as already described, and then enzymatic—to reduce the molecular weight followed by (2) metabolisation of the chain monomers by the microorganisms present [5,13,158]. As in the case of water uptake, microbial communities target predominantly the amorphous regions, since the polymer chains are more flexible there, increasing the crystallinity of the attacked regions and, thus, embrittling the polymer matrix [13,191,192,220].

Finally, in PLA matrix composites, the presence of the fillers usually favours moisture uptake and, hence, the decomposition by microorganisms [5,212], which, besides affecting the total mass as in [208], may also impact the overall mechanical, thermal and surface properties. Alterations in the microstructure, promoting embrittlement, and, with them, an increase in the glass transition temperature and crystallinity degree have been reported in the literature [5,212].

### 10.2. Polyhydroxyalkanoates

PHA—their most important representative being poly(hydroxybutyrate) (PHB)—are biobased and biodegradable polymers produced by soil bacteria, typically stiff and brittle, with low permeability to water and oxygen [13,25]. Taking a closer look at PHB, its microstructure consists of alternating layers of crystalline and amorphous domains, the latter being preferentially targeted by hydrolytic degradation.

#### 10.2.1. Water

PHB is known to degrade in water, with the tensile strength decreasing significantly in the initial stage of degradation and slowing down as decomposition evolves and the rate at which decomposition occurs depending on the water inorganic composition and temperature [208,222]. Mass losses of 7% and 35% were recorded after 180 and 358 days of water exposure, respectively [223], and a value of 8.5% was measured for simulated conditions over one year [13,185].

#### 10.2.2. Temperature

Temperature is an important factor when assessing the biodegradability of PHB composites [25], e.g., samples at 40 ${}^{{}^{\circ}}$C exhibited a weight loss of 0.64% per day [224], whilst a 50% weight loss was registered for specimens kept at temperatures of up to 30 ${}^{{}^{\circ}}$C during 50 days.

#### 10.2.3. Seawater

PHB is totally biodegradable in seawater, experiencing, much unlike PLA, fast hydrolysis [11]; several studies have been reviewed in [11]. Surface erosion was generally considered the main initiator of degradation, promoted by the high availability of microorganisms capable of degrading these matrices—even if still slower than in compost, after which other mechanisms would act in the polymer matrix. Different studies have been performed with rather distinct outcomes: weight loss of 7% after 1 year at 25 ${}^{{}^{\circ}}$C [185] (Figure 26) against 42% after just 160 days at 29 ${}^{{}^{\circ}}$C or even 60% after 35 days in [11,13]. One common observation, nevertheless, is that the degradation of the polymer chains is uniform and independent of the chain length [13]. Furthermore, after 140-day submergence in a marine environment, the degradation levels of both the amorphous and the crystalline phases were reportedly the same [225,226]. Finally, PHA fibres, albeit not exhibiting signs of hydrolysis even after 120 days exposed to a saline solution, contrasting against PHB/PHV ones, have been shown to affect the degradability of the polymer matrix itself due to their water uptake [227].

#### 10.2.4. Microorganisms

Comparatively to seawater, these polymers degrade much faster in soil and compost [11] according to multiple studies [5,229,230,231,232,233,234]. The involved kinetics depend strongly on temperature, as it influences the biotic activity, leading to distinct rates over a period of 200 days, namely 0.05%/day at 15 ${}^{{}^{\circ}}$C, 0.12%/day at 28 ${}^{{}^{\circ}}$C and 0.45%/day at 40 ${}^{{}^{\circ}}$C, as reported by [13]. Likewise, a mass loss of 98% after one year has been measured by [13]. These results have been replicated for composting conditions [13], with results ranging from a mass loss of 6% for temperatures of 6–32 ${}^{{}^{\circ}}$C over 150 days all the way to complete degradation at 58 ${}^{{}^{\circ}}$C after 28 days. Degradation occurs preferentially in the amorphous regions, being accelerated by applied strains (both static, creep-inducing, and cyclic), inducing a change in semi-crystalline domains and increasing the amorphous fraction [235]. Finally, regarding composites, the incorporation of TiO${}_{2}$ particles has been reported to slow the degradation under sediment burial [232].

### 10.3. Thermoplastic-Starch

TPS is a biobased and biodegradable polymer(-matrix) boasting an amorphous structure that is synthesised by adding natural plasticisers to starch obtained from agricultural resources—such as potatoes, corn and rice, all rich in polysaccharides—leading to a thermoplastic polymer [13,25]. Typically, TPS-based composites exhibit low mechanical strength and impact resistance, being typically brittle and affected by water uptake [236]. Regarding pure mechanical fatigue, it has been shown that reinforcements, particularly fibres, have no effect on the failure mechanisms, which, again, confirms our assumption that the fatigue behaviour for TPS-based composites depends solely on the matrix [162]. Degradation results under UV radiation, moisture, temperature and microbial attack range from mass loss to surface oxidation, including changes in tensile and thermal properties [5,237].

#### 10.3.1. Water

Starch alone possesses many polar groups that react forming hydrogen bonds, leading to high water absorption [13]. This, nonetheless, can be offset by different additives, both in the neat polymer and as reinforcements in composites [25]. Looking specifically at composites, cellulose fibres have been recorded to enhance both mechanical and thermal properties, whilst decreasing the water uptake [238,239].

#### 10.3.2. Seawater

TPS has proven to be degradable in seawater, albeit slowly, with several blends and composites registering weight loss and deterioration of tensile properties almost exclusively in the starch fraction [240,241], as occurred in, e.g., PHB/TPS blends immersed in coastal water in Puerto Rico over a period of 1 year or PE/TPS blends exposed in laboratory tests simulating harbour conditions in the Baltic Sea over 20 months [240,242,243]. Enzymatic hydrolysis has been reported to be the main contributing mechanism, which, in composites, is favoured by the increased water uptake associated with most reinforcements, and by high levels of strain that augment the surface area, thus helping the microbial communities in seawater to infiltrate the composite. Moreover, as expected, the rate of degradation depends on the microorganisms present: in river water (highest number of bacteria), a weight loss of 32% was reported, decreasing to 3.3% in a marine environment, with studies indicating rates as high as 2% day${}^{-1}$ [11].

#### 10.3.3. Microorganisms

Starch and TPS composites are known to degrade quickly under the influence of biotic factors, with studies performed both in soil burial and composting conditions [244,245,246,247,248]. Regarding the former, values of 100% and 72.6% mass loss have been reported in the literature for a period of 30 days at 30 ${}^{{}^{\circ}}$C [249] and 25 ${}^{{}^{\circ}}$C [250], respectively, whereas for the latter, an unexpected longer time was measured in several studies [251,252], taking up to 84 days at 60 ${}^{{}^{\circ}}$C to fully decompose. A low starch content is, additionally, prone to slowing fatigue crack propagation and diminishing the incidence of microbial attack, with a high starch content promoting faster degradation and creep kinetics due to an overall increase in the penetration of water, light, oxygen and microbial attack. Contrasting results have, nevertheless, been found in the literature, e.g., with studies reporting a long time for complete degradation—as long as 12 years for some TPS/PE blends under composting conditions [25,180].

## 11. Conclusions

This work reviews the physical and chemical phenomena involved in the environmental fatigue and creep performance of polymer-matrix composites, relating them to different microstructures, either semi-crystalline or amorphous. The studies mentioned throughout this work are gathered in Table A1, Table A2 and Table A3 in Appendix A, wherein we collect the specimen dimensions, the testing conditions and duration and the most relevant findings.

We begin by describing how solvents, particularly water, induce swelling, which may lead to hydrolysis and changes in $T_{g}$ over enough time. At low temperatures, water stiffens a polymer matrix, promoting earlier fatigue failure for high loads, whereas, above $T_{g}$, it acts as a plasticising agent, increasing the strains required to initiate a crack, a behaviour clearly established for, e.g., PP and PA composites.

The effects of seawater are also discussed, stemming from a combination of mechanical (currents); biotic (bacteria and fungi); chemical (pH and salinity); and thermal factors (such as the temperature profile of the ocean). All of these contribute to shortening the service life of a PMC, since under creep and/or fatigue loads, its degradation rate is accelerated—which occurs, e.g., for PHB matrices. In addition, the type and content of the reinforcements may increase the fatigue damage kinetics, particularly if fibres suffer water absorption or biodecomposition.

On the topic, biotic agents promote chemical degradation, as their enzymatic action can metabolise some polymers and, particularly, natural fillers, leading to a change in the structure and/or composition, whose products can accelerate the loss of integrity. This has been shown to be particularly relevant for natural fibres and/or biodegradable composites, such as those with PLA and TPS matrices, especially under tensile fatigue/creep loads that increase the exposed area.

Focusing on liquid chemical agents, it was found that penetration is paramount for a PMC. Permeation initially induces chemical reactions that dissolve the matrix. Further propagation leads to the destruction of the interface bonds, and fibres may even be left exposed to the solution resulting in surface cracks. Fatigue and creep responses are thus reduced by up to 58% in the slope of the *S*-*N* curves due to a drop in the interface properties and ability of the matrix to transfer loads between fibres, a behaviour clearly established for, e.g., vinyl ester.

Another important factor is the incidence of UV radiation, particularly UV-B, which is powerful enough to break the chemical bonds within polymer chains in a matrix, particularly at a thin layer on the surface, embrittling this region and, upon fatigue loads, promoting earlier cracking and further embrittlement as the exposed area increases—for instance, in PLA composites.

Temperature was found to be one of the most important environmental factors related to the decrease in fatigue and creep resistance. Higher temperatures lead to the movement and sliding of polymeric chains, primarily in amorphous regions, leading to a decrease in stiffness and, thus, to the reduction of fatigue and creep resistance. Differences between the coefficients of thermal expansion of the matrix and reinforcements are the source of many other effects, such as the development of internal stresses when exposed to cyclic thermal variations or cryogenic temperatures, ultimately causing the development of micro-cracks and fibre–matrix debonding, further reducing fatigue performance due to accelerated crack propagation—as seen in epoxi-matrix composites.

The assessment of lighting strike influence, an important factor for open field use composites, was also conducted in this work. The creation of the high current channel coupled with the low electrical conductivity of the matrix creates a very high temperature gradient near the attachment point, originating small interlaminar spaces in the process. The accumulation of pyrolysis gases inside the gaps induce an explosive effect which extensively damages the composite and consequently has its fatigue and creep resistance reduced.

Particularly evident in the case of biodegradable and bio-based polymer matrices, the available literature on fatigue and creep of PMCs focuses mostly on the assessment of the mechanical properties under constant laboratory conditions, using accelerated testing methods and curve-fitting to extrapolate the full behaviour. Nevertheless, the results gathered from such tests may not be directly extrapolated, due to unforeseen alterations in behaviour that the short duration of the tests fails to capture. Not only are longer tests required, but also few studies follow the existing testing standards, preventing direct comparison of the obtained results. Still, the different environmental conditions must be assessed, as this area remains to be thoroughly explored. Most works available focus on epoxy-matrix composites, with thermoplastics only recently coming under the spotlight. Regarding biodegradable and biobased polymers, pure mechanical fatigue and creep studies are even rarer, with almost all of the available literature studying degradation and decomposition effects, mostly from a chemical or environmental standpoint, and not really concentrating on the mechanical properties that are the focus here. These current trends as well as opportunities and challenges for further studies are compiled in Table A4 in Appendix B for each polymer. Beyond these, other limitations include the existence of size factors that affect crack nucleation and propagation, which have yet to be thoroughly explored beyond the fracture mechanics field, accounting for the environmental factors mentioned. Similarly, the manufacturing method plays an important role in determining the properties of a given composite and is left to be tackled in coming studies. Future studies should focus, on the one hand, on the development of the required methods, and, on the other hand, on the actual creep and fatigue properties of such polymer matrices, in an effort to keep up with the new composites that are being developed, eventually contributing to understanding the complex interactions occurring between the different mechanisms, while allowing for tailoring of the relevant engineering properties.

## Figures and Tables

**Figure 1 materials-16-03913-f001:**
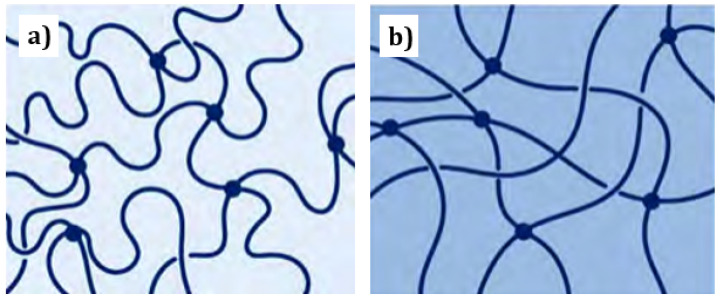
(**a**) Dry polymer chains; (**b**) swollen polymer chains. Adapted with permission from Zhao et al. [3]. Copyright 2021 American Chemical Society.

**Figure 2 materials-16-03913-f002:**
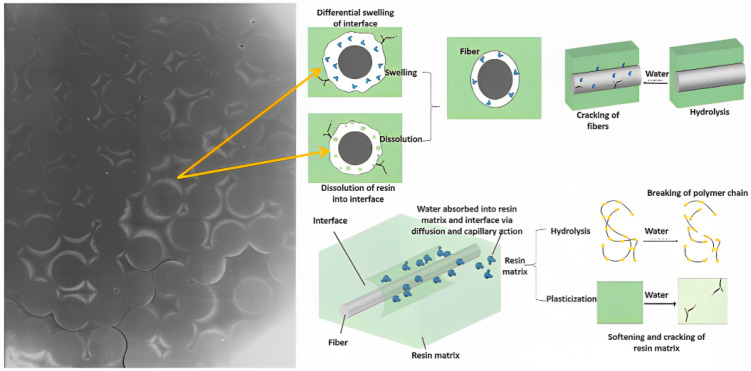
Moisture absorption in fibre-reinforced PMCs. Adapted with permission from Weitsman [8]. Copyright 2006 Elsevier. Adapted with permission from Liu et al. [10]. Copyright 2020 Elsevier.

**Figure 3 materials-16-03913-f003:**
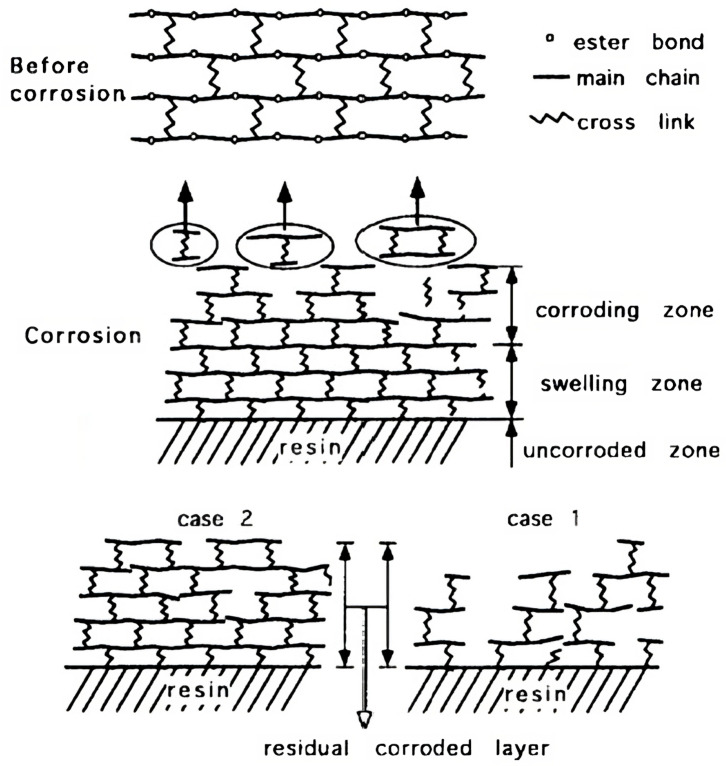
Corrosion process of corroded layer forming. Reprinted with permission from Hojo et al. [19]. Copyright 1998 Springer Nature.

**Figure 4 materials-16-03913-f004:**
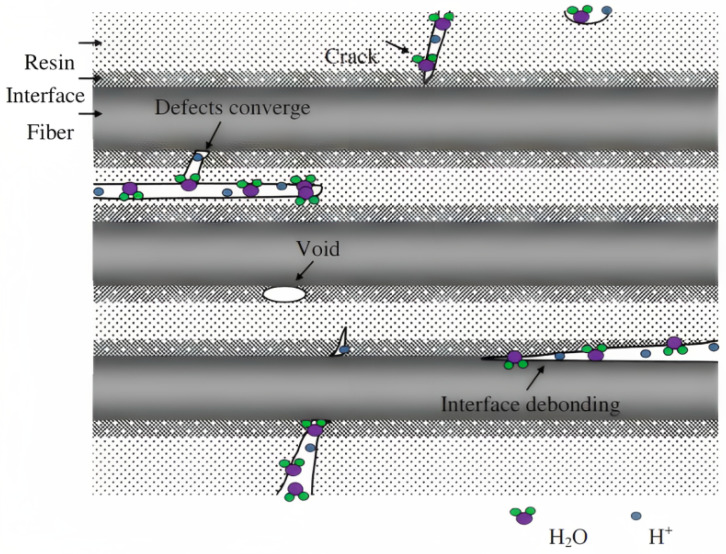
Corrosion mechanisms under exposure to an acidic solution. Reprinted with permission from Wang et al. [18]. Copyright 2018 American Society of Civil Engineers.

**Figure 5 materials-16-03913-f005:**
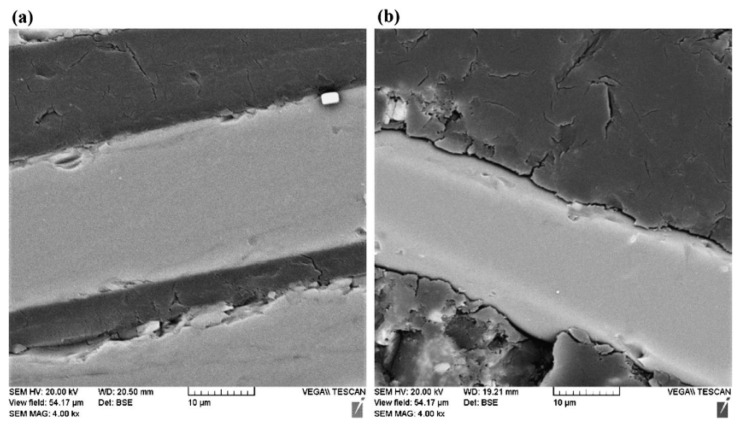
Interface damage on E-glass/PE: (**a**) before exposure; (**b**) after exposure for 4 weeks at 70 ${}^{{}^{\circ}}$C in a 10 wt.% HCl solution. Reprinted with permission from Amini and Khavandi [21]. Copyright 2019 Springer Nature.

**Figure 6 materials-16-03913-f006:**
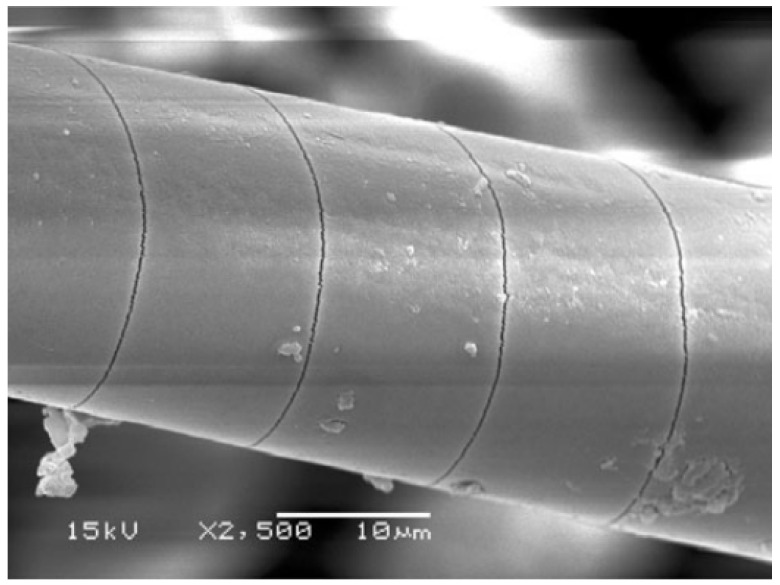
Spiral crack of a glass fibre after exposure to an acidic solution. Reprinted with permission from Hammami and Al-Ghuilani [24]. Copyright 2004 John Wiley and Sons.

**Figure 8 materials-16-03913-f008:**
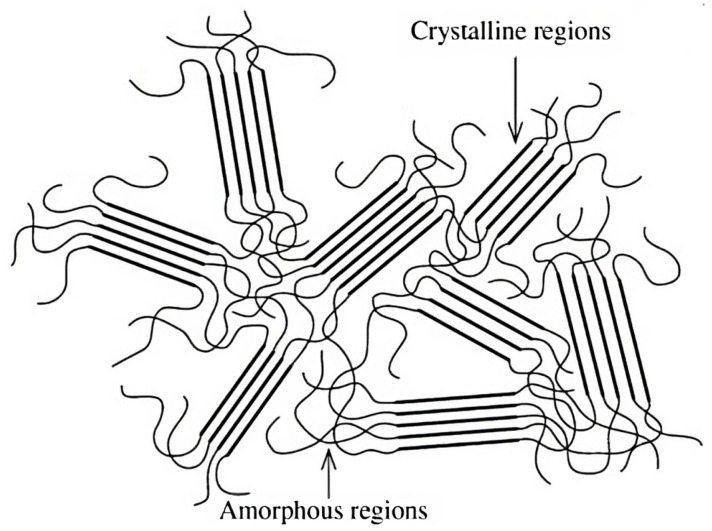
Representation of crystalline and amorphous regions in semicrystalline plastics. Reprinted with permission from McKeen [33]. Copyright 2008 Elsevier.

**Figure 9 materials-16-03913-f009:**
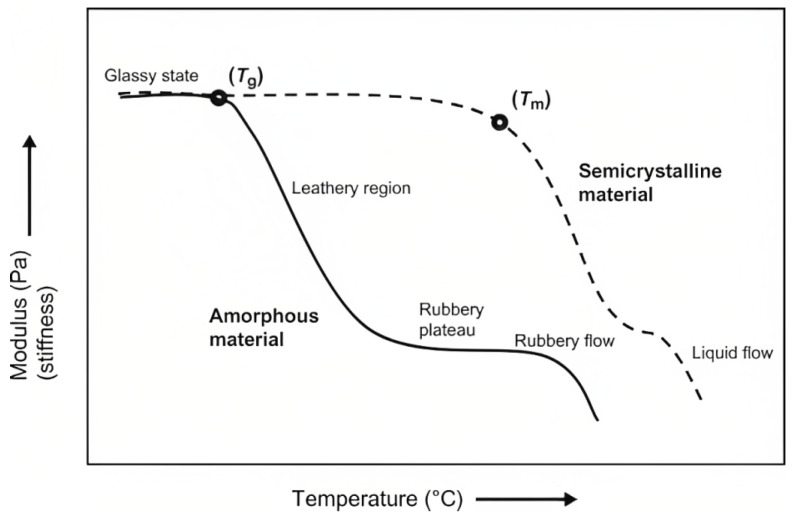
Melting and softening behaviour of semicrystalline and amorphous materials. Reprinted with permission from Shrivastava [32]. Copyright 2018 Elsevier.

**Figure 10 materials-16-03913-f010:**
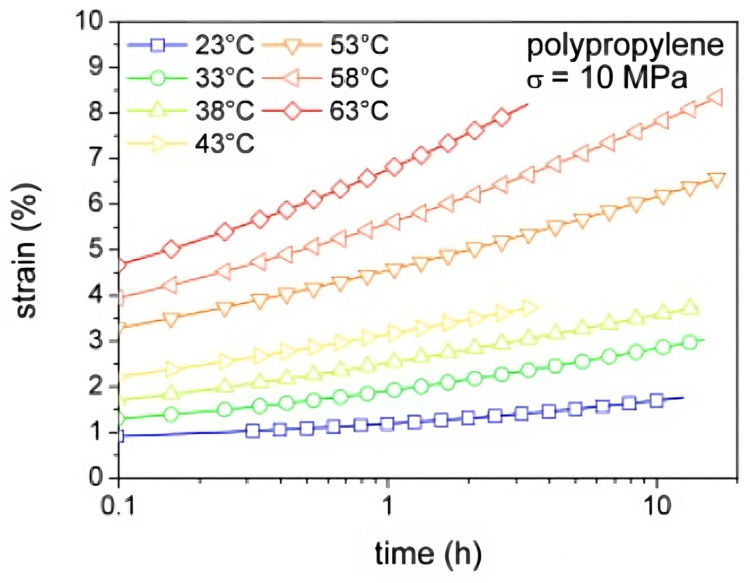
Temperature creep curves for polypropylene under a tensile load of 10 MPa. Reprinted with permission from Achereiner et al. [37]. Copyright 2013 Elsevier.

**Figure 11 materials-16-03913-f011:**
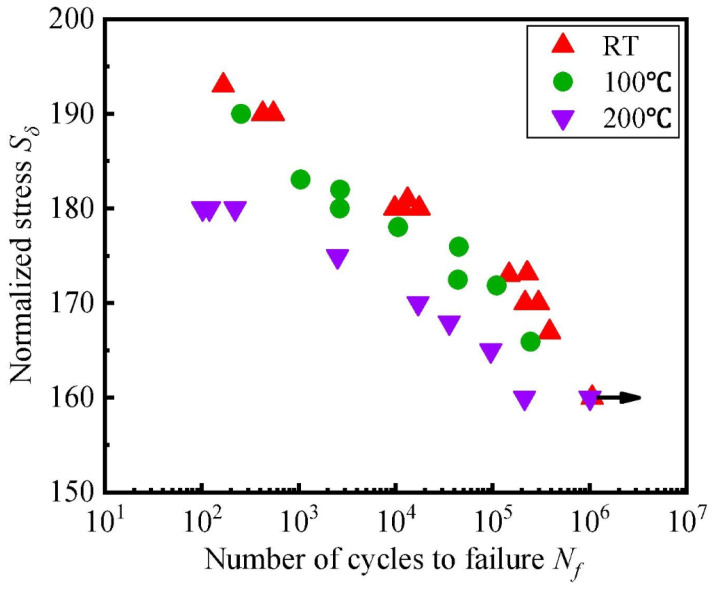
*S*-*N* curves for three different temperatures. Reprinted with permission from Guo et al. [40]. Copyright 2023 Elsevier.

**Figure 12 materials-16-03913-f012:**
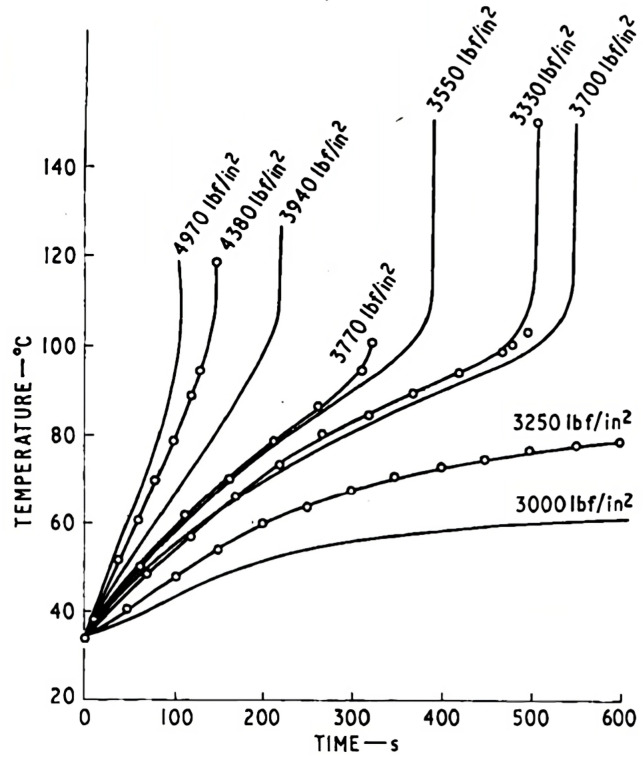
Experimental temperature rise curves in polymethylmethacrylate specimens, 3000 rev/min at 25 °C. Reprinted with permission from Constable et al. [42]. Copyright 1970 SAGE Publications.

**Figure 13 materials-16-03913-f013:**
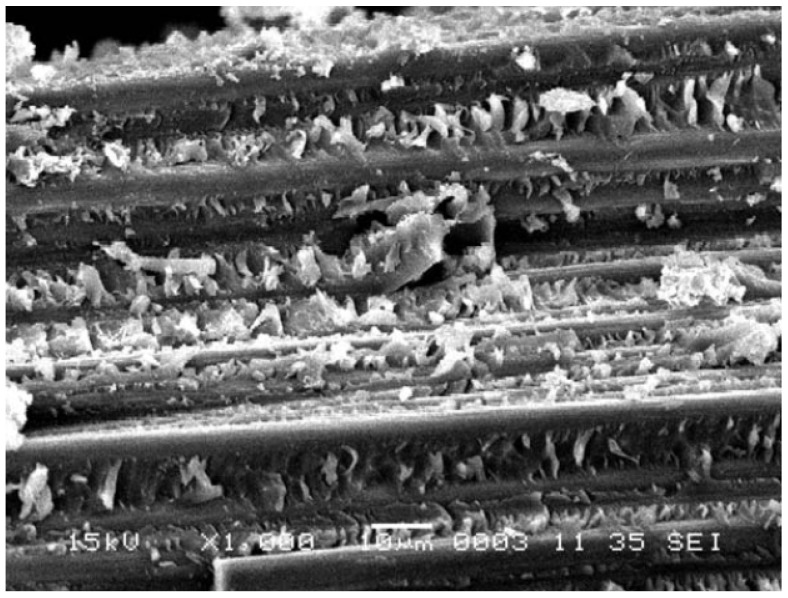
Micrograph showing matrix crackings and delamination of a cryogenically conditioned carbon/epoxy specimen. Reprinted with permission from Kumar et al. [44]. Copyright 2009 SAGE Publications.

**Figure 14 materials-16-03913-f014:**
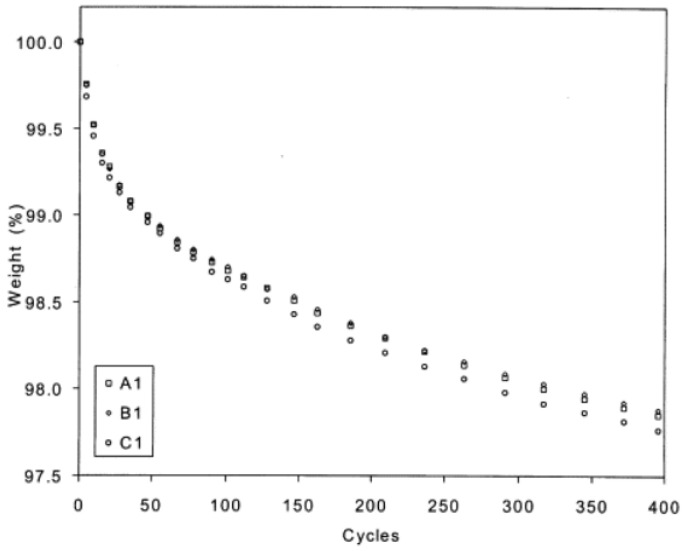
Effect of a number of thermal cyclings on a carbon fibre prepreg sample weight change. Reprinted with permission from Chung et al. [50]. Copyright 2000 Elsevier.

**Figure 15 materials-16-03913-f015:**
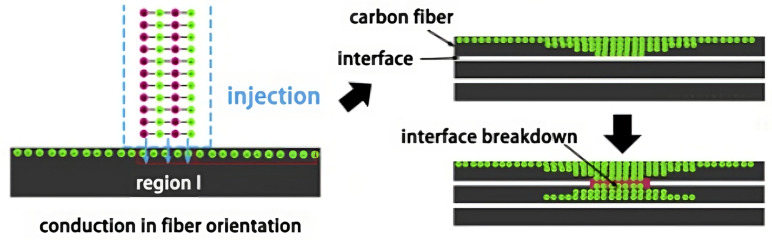
Schematic illustration of the lightning process on a CFRP composite. Reprinted with permission from Wang et al. [55]. Copyright 2020 Elsevier.

**Figure 16 materials-16-03913-f016:**
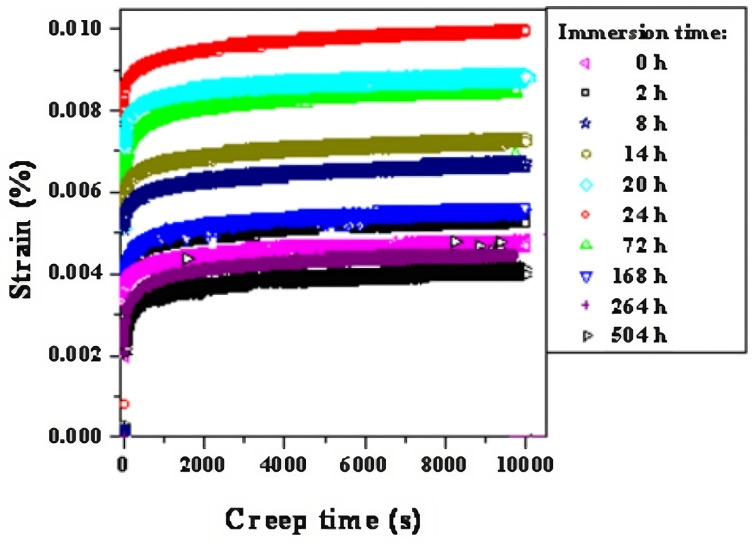
Effect of water uptake on creep behaviour of glass/epoxy specimens. Reprinted with permission from Papanicolaou et al. [69]. Copyright 2009 Taylor and Francis.

**Figure 17 materials-16-03913-f017:**
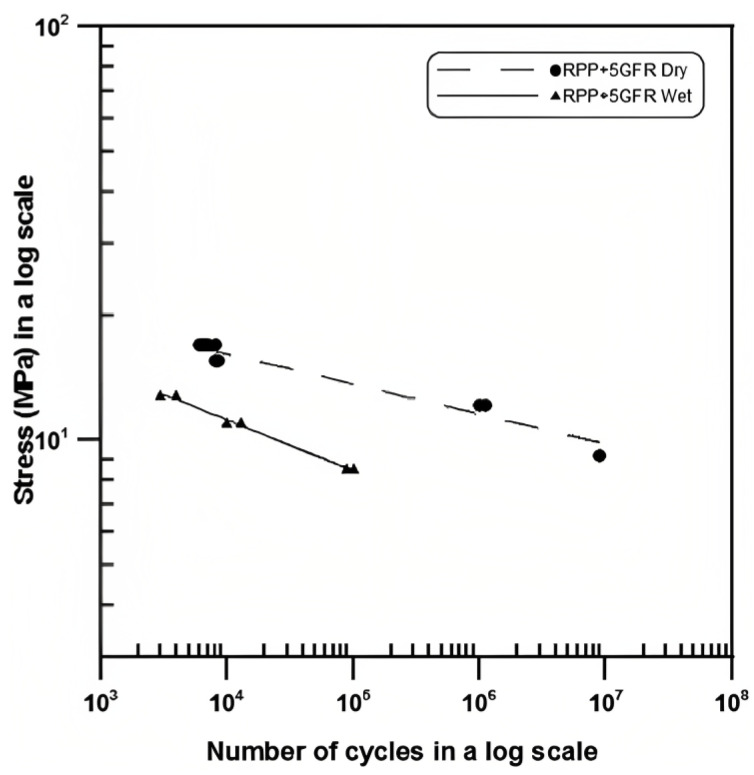
Effect of moisture on fatigue behaviour of polypropylene reinforced with 5 wt% of short glass fibre. Reprinted with permission from Abdelhaleem et al. [109]. Copyright 2018 SAGE Publications.

**Figure 18 materials-16-03913-f018:**
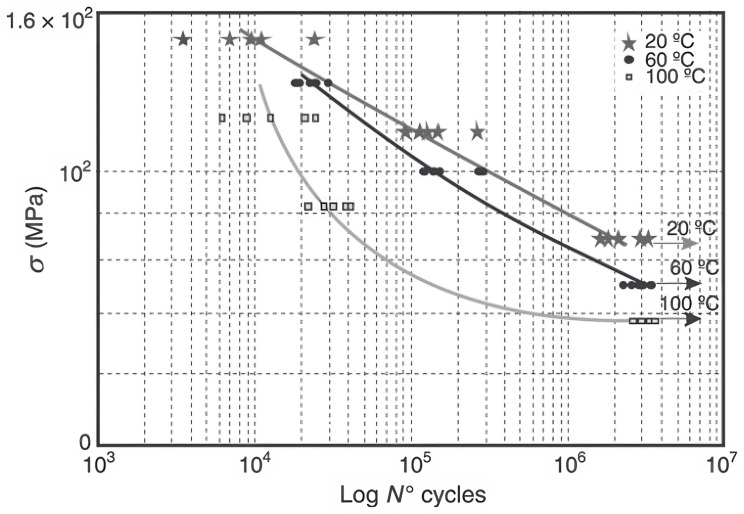
Effect of temperature on the fatigue life of glass fibre-reinforced PP. Reprinted with permission from Viña et al. [111]. Copyright 2011 John Wiley and Sons.

**Figure 19 materials-16-03913-f019:**
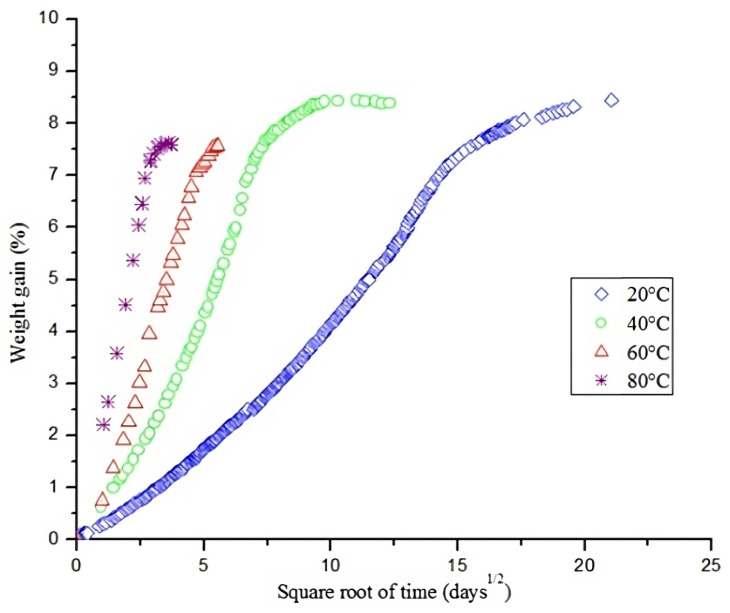
Effect of temperature on the moisture uptake of a hemp/polypropylene composite. Reprinted with permission from Toubal et al. [115]. Copyright 2016 John Wiley and Sons.

**Figure 20 materials-16-03913-f020:**
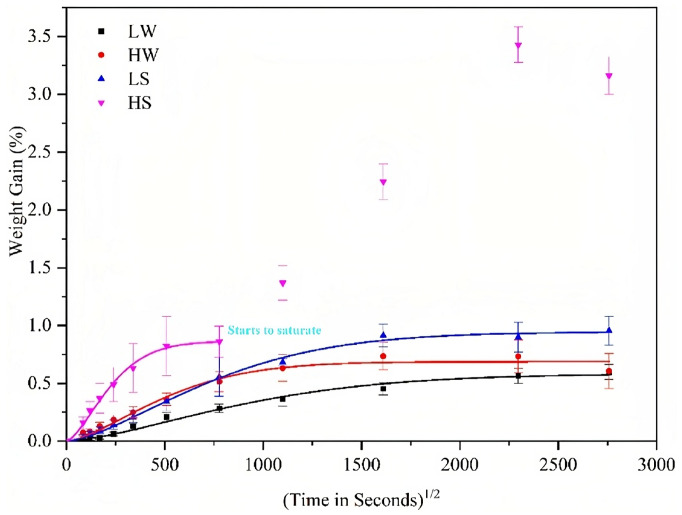
Water uptake of GFRP sheets under different environments (LW and HW: distilled water at 25 and 60 ${}^{{}^{\circ}}$C; LS and HS: seawater at 25 and 60 ${}^{{}^{\circ}}$C). Reprinted with permission from Dong et al. [116]. Copyright 2022 Elsevier.

**Figure 21 materials-16-03913-f021:**
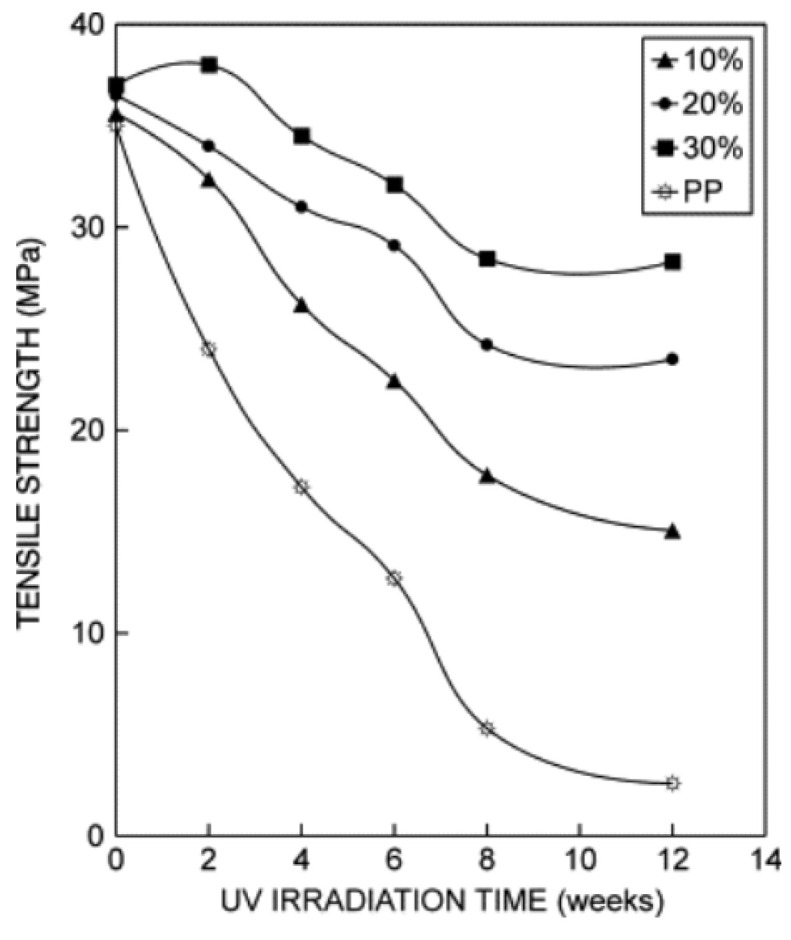
Variation of tensile strength of sisal/PP composites when exposed to UV radiation for different periods. Reprinted with permission from Joseph et al. [118]. Copyright 2002 Elsevier.

**Figure 22 materials-16-03913-f022:**
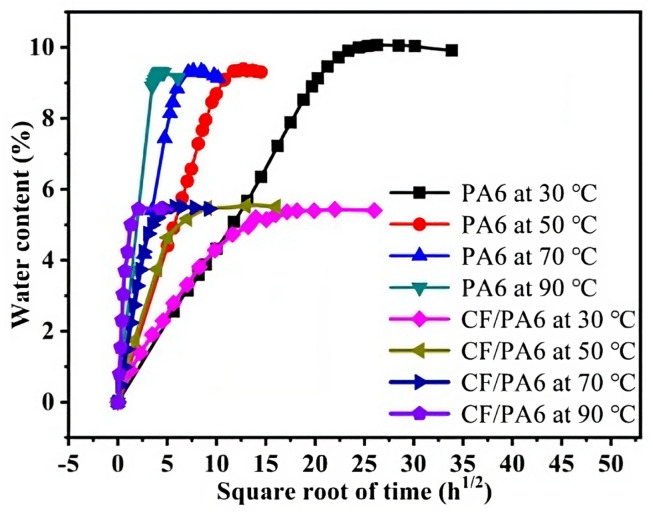
Effect of temperature on water uptake of PA6 and CF/PA6. Reprinted with permission from Lei et al. [132]. Copyright 2021 Elsevier.

**Figure 23 materials-16-03913-f023:**
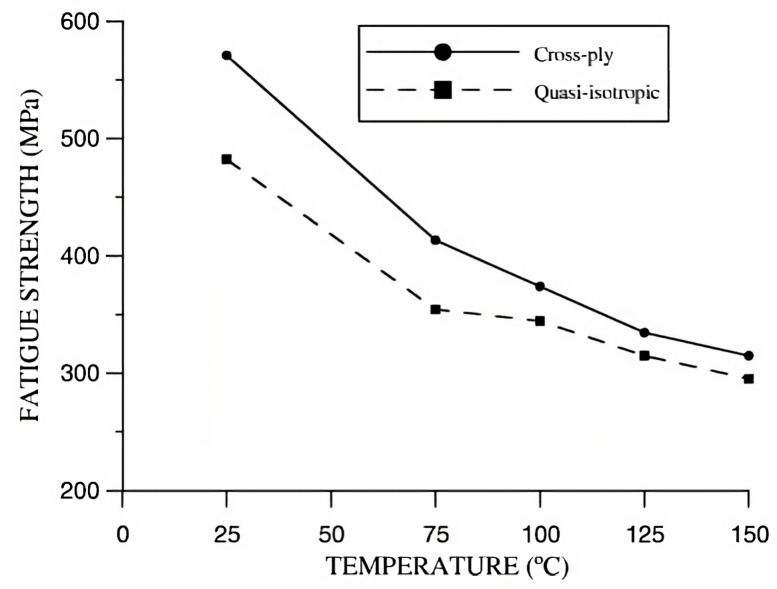
Effect of temperature on the fatigue life of cross-ply and quasi-isotropic AS-4/PEEK laminates. Reprinted with permission from Jen et al. [140]. Copyright 2008 Elsevier.

**Figure 24 materials-16-03913-f024:**
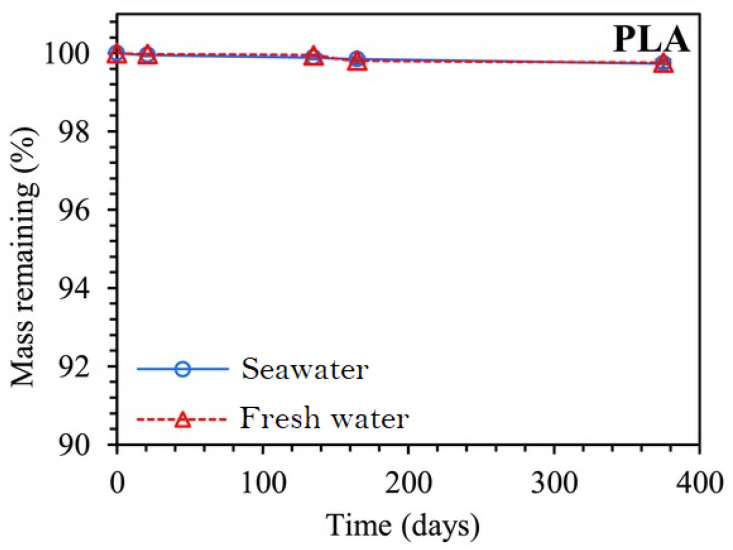
Mass loss for PLA samples in seawater and fresh water. Reprinted with permission from Bagheri et al. [185]. Copyright 2017 John Wiley and Sons.

**Figure 25 materials-16-03913-f025:**
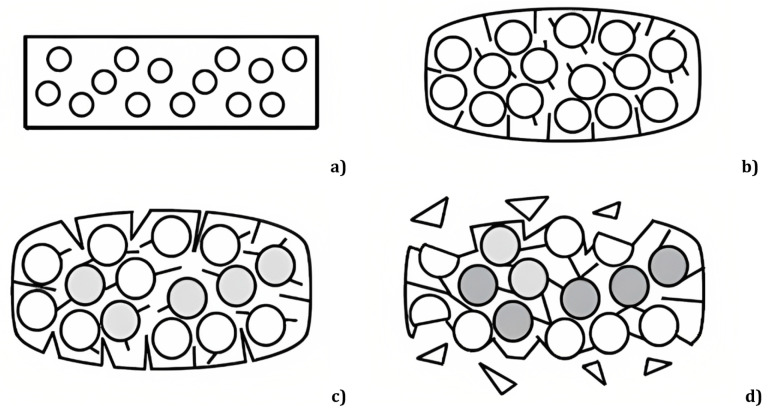
Stages of a PLA composite degradation: (**a**) initial material; (**b**) cracks appear and the contact area increases with increased enzymatic degradation of the fibres; (**c**) hydrolytic degradation of the PLA matrix; (**d**) embrittlement and disintegration. Adapted with permission from Bayerl et al. [221]. Copyright 2014 Elsevier.

**Figure 26 materials-16-03913-f026:**
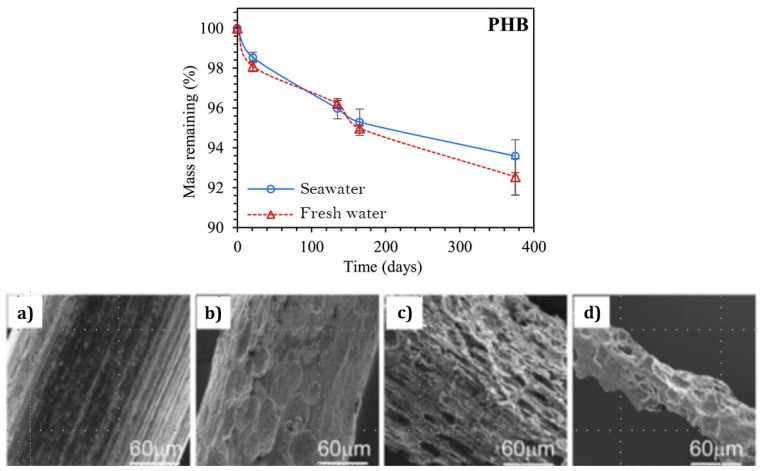
(**Top**) Mass loss for PHB samples in seawater and fresh water. Reprinted with permission from Bagheri et al. [185]. Copyright 2017 John Wiley and Sons. (**Bottom**) PHB samples in seawater over 12 months: (**a**) not soaked; (**b**–**d**) soaked in different marine environments. Adapted with permission from Sekiguchi et al. [228]. Copyright 2011 Elsevier.

## Data Availability

No new data were created or analyzed in this study. Data sharing is not applicable to this article.

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
