# Peer review of "Polymer-Matrix Composites: Characterising the Impact of Environmental Factors on Their Lifetime"

_materials, 2023, doi:10.3390/ma16113913_

Round 1
Reviewer 1 Report
The manuscript was well worked, the information carefully selected and presented, however some typographical errors were detected in lines 942, 945 and 1001, the symbol for the temperature in °C is missing. References 39 and 198 are incomplete.
No problems or inconsistencies with the English language were detected.
Author Response
The manuscript was well worked, the information carefully selected and presented, however some typographical errors were detected in lines 942, 945 and 1001, the symbol for the temperature in °C is missing. References 39 and 198 are incomplete.
R: Suggestions accepted. References 39 and 198 (now 200) were corrected as well as the typographical errors (lines 952, 955 and 1011).
Reviewer 2 Report
1. Illustration should be standardized in this study. for example, When multiple images appear, please use a), b),.., unified annotations.
2. Suggestions should be supplied for future research in the abstract.
3. This study was summarized from two parts: environmental factors and representative polymers, the conclusion of this study should also be summarized in two parts as the environment and representative compounds. In addition, the conclusion should be presented in sections.
4. The impact mechanism of seawater on the polymers should be supplemented with more detailed information. Alternatively, please merge the seawater effect with the water effect.
5. Water and moisture should be modified to water, as moisture also belongs to water.
6. Microorganisms-biological factors should be modified microorganisms.
Author Response
1.Illustration should be standardized in this study. for example, When multiple images appear, please use a), b),.., unified annotations.
R: Suggestion accepted.
2.Suggestions should be supplied for future research in the abstract.
R: Suggestion accepted – lines 17-18: “Future studies should focus on other matrices beyond epoxy as well as on the development of standardised testing methods, allowing for a direct comparison of the experimental results obtained in different conditions.”.
3.This study was summarized from two parts: environmental factors and representative polymers, the conclusion of this study should also be summarized in two parts as the environment and representative compounds. In addition, the conclusion should be presented in sections.
R: We thank the reviewer for the suggestion; however, according to the guidelines for authors, the conclusion cannot be subdivided into different parts and/or sections. Still, to reply to the reviewer’s comment, we have organized this section into several paragraphs, each one corresponding to a specific part and/or section of the main text.
- Part I, section 1.1 – conclusions are drawn in lines 1125-1129, featuring examples from Part II
- Part I, section 1.2 – conclusions are drawn in lines 1130-1136, featuring examples from Part II
- Part I, section 2 – conclusions are drawn in lines 1143-1149, featuring examples from Part II
- Part I, section 3 – conclusions are drawn in lines 1137-1142, featuring examples from Part II
- Part I, section 4 – conclusions are drawn in lines 1150-1154, featuring examples from Part II
- Part I, section 5 – conclusions are drawn in lines 1155-1163, featuring examples from Part II
- Part I, section 6 – conclusions are drawn in lines 1164-1169, featuring examples from Part II
- Lines 1170-1194 sum up Part II and reflects on the future studies required to provide further insight into the topic.
4.The impact mechanism of seawater on the polymers should be supplemented with more detailed information. Alternatively, please merge the seawater effect with the water effect.
R: Suggestion accepted. We have merged the “Seawater” section with the one on “Solvents – esp. water” together in the “Solvents” section, that now features 2 subsections – lines 101-203.
5.Water and moisture should be modified to water, as moisture also belongs to water.
R: Suggestion accepted.
6.Microorganisms-biological factors should be modified microorganisms.
R: Suggestion accepted.
Reviewer 3 Report
This review on the influence of various environments on the change in the properties of polymer composites is relevant and useful to a wide range of researchers. The authors, on the one hand, briefly, and on the other hand, quite informatively explained the influence and causes of environmental influence on the degradation of the properties of both polymer matrices and reinforcing components. It should also be noted the good presentation of the material, namely the division of the article into two parts. In the first part, the authors consider the general effect of external influences on the degradation of properties in polymer composites, and the second part is devoted to a more detailed description of the influence of external influences on specific types of composites. This division, in my opinion, allowed the authors to make their review clear, compact and informative. Of course, this article and the work of the authors deserves a positive assessment. Nevertheless, there were inaccuracies in this article, which are listed below.
1. From lines 88-92, it can be understood that the absorption of water leads to an increase in the glass transition temperature. While the opposite is indicated in lines 108-111, that is, the absorption of water by the polymer reduces the glass transition temperature.
2. Lines 390-392. It is not clear how Figure 9 symbolizes the effect of stiffens and strength on fatigue. This figure shows only the effect of temperature on the elastic modulus.
Author Response
This review on the influence of various environments on the change in the properties of polymer composites is relevant and useful to a wide range of researchers. The authors, on the one hand, briefly, and on the other hand, quite informatively explained the influence and causes of environmental influence on the degradation of the properties of both polymer matrices and reinforcing components. It should also be noted the good presentation of the material, namely the division of the article into two parts. In the first part, the authors consider the general effect of external influences on the degradation of properties in polymer composites, and the second part is devoted to a more detailed description of the influence of external influences on specific types of composites. This division, in my opinion, allowed the authors to make their review clear, compact and informative. Of course, this article and the work of the authors deserves a positive assessment. Nevertheless, there were inaccuracies in this article, which are listed below.
R: Thank you very much for your positive feedback.
- From lines 88-92, it can be understood that the absorption of water leads to an increase in the glass transition temperature. While the opposite is indicated in lines 108-111, that is, the absorption of water by the polymer reduces the glass transition temperature.
R: Suggestion accepted and corrected accordingly. We thank the reviewer for the suggestion. The text has been clarified - see changes in lines 92, 111-112.
- Lines 390-392. It is not clear how Figure 9 symbolizes the effect of stiffens and strength on fatigue. This figure shows only the effect of temperature on the elastic modulus.
R: Suggestion accepted. We agree with the reviewer and we adapted the text accordingly – see lines 384-391, wherein we mention Figure 9, and lines 392-398.
Reviewer 4 Report
The review article brings knowledge in polymer-matrix composites and their environmental assessment. The reviewer found many of the arguments repetitive throughout the paper, most of the results and test methods are similar and conversation revolved around the same. There are specific concerns, as follows:
1. is there a size factor considered when evaluating the crack propagation?
2. additive manufacturing is an advanced technology; however, the paper contains very little additive manufacturing validation other than focusing more on the conventional manufacturing testing method.
3. It is recommended that the authors be specific about the code or standard use other than mentioning the test name, like salt water test and tested conditions.
4. Do all the characterizing methods have a respective predictive model? Reviewing the paper seems like most of the characterizing methods are based on experimental methods.
5. The paper reviews the physical and chemical phenomena causing fatigue and creep, is surface pitting due to impurities considers one of the physical phenomena?
6. Temperature is discussed in the paper lots of times, but it is not clear how the temperature change in the real world like a drastic change of temperature from hot to cold or vice versa would have effect on any of the polymers, matrix, and composites.
7. Line 450; Could this heat-absorbing problem be tackled by adding electric conductive materials into the polymer? The electric current will be transported to storage when the lighting strikes the PMC.
The language needs revision/editing.
Author Response
The review article brings knowledge in polymer-matrix composites and their environmental assessment. The reviewer found many of the arguments repetitive throughout the paper, most of the results and test methods are similar and conversation revolved around the same. There are specific concerns, as follows:
- is there a size factor considered when evaluating the crack propagation?
R: We agree with the reviewer’s comment, but this size factor is usually assessed in fracture mechanics, not on the environmental-related literature that was gathered in this review. Indeed, this size factor was not mentioned in the reviewed (experimental) literature. As such, this limitation is now included in the conclusion (lines 1185-1188).
- additive manufacturing is an advanced technology; however, the paper contains very little additive manufacturing validation other than focusing more on the conventional manufacturing testing method.
R: We thank the reviewer for the suggestion. In fact, this comment is particularly relevant for PLA, being one of the most common polymers employed in additive manufacturing. Nevertheless, the majority of the degradation studies still focus on conventionally manufactured samples; only recent studies mention engineering applications of PLA, with only a few of them including additive manufacturing. Additionally, this review is focused on listing degradation mechanisms and providing significative examples for important engineering polymers, not covering the manufacturing method, identified as a limitation in lines 1188-1189. However, we would like to thank the reviewer for the interesting idea for a new study.
- It is recommended that the authors be specific about the code or standard use other than mentioning the test name, like salt water test and tested conditions.
R: Suggestion accepted. We added the pertaining testing standard whenever possible. Additionally, since some of the studies mentioned did not employ normalised testing standards, this was added as a limitation in lines 1175-1177.
Please see changes in lines:
- Appendix A.1., Table A.1., Polymer Epoxy; Hygrothermal effect, Reference 78
- Appendix A.1., Table A.1., Polymer Epoxy; Hygrothermal effect, Reference 80
- Appendix A.1., Table A.1., Polymer Epoxy; Hygrothermal effect, Reference 84
- Appendix A.1., Table A.1., Polymer Epoxy; Seawater effect, Reference 86
- Appendix A.1., Table A.1., Polymer Epoxy; Seawater effect, Reference 87
- Appendix A.1., Table A.1., Polymer Epoxy; Liquid chemical agents effect, Reference 88
- Appendix A.1., Table A.1., Polymer Epoxy; Radiation effect, Reference 91
- Appendix A.1., Table A.1., Polymer Vinyl ester; Seawater effect, Reference 94
- Appendix A.1., Table A.1., Polymer Vinyl ester; Seawater effect, Reference 96
- Appendix A.1., Table A.1., Polymer Vinyl ester; Seawater effect, Reference 97
- Appendix A.1., Table A.1., Polymer Polyester; Seawater effect, Reference 101
- Appendix A.2., Table A.2., Polymer PP; Water effect, Reference 108
- Appendix A.2., Table A.2., Polymer PP; Water effect, Reference 252
- Appendix A.2., Table A.2., Polymer PP; Water effect, Reference 253
- Appendix A.2., Table A.2., Polymer PP; Seawater effect, Reference 115
- Appendix A.2., Table A.2., Polymer PP; Seawater effect, Reference 116
- Appendix A.2., Table A.2., Polymer PA; Water effect, Reference 130
- Appendix A.2., Table A.2., Polymer PA; Water effect, Reference 131
- Appendix A.2., Table A.2., Polymer PEEK; UV radiation effect, Reference 150
- Appendix A.2., Table A.2., Polymer PEEK; UV radiation effect, Reference 151
- Appendix A.3., Table A.3., Polymer PLA; Microorganisms effect, Reference 157 – entry 1 and 2
- Appendix A.3., Table A.3., Polymer PLA; Microorganisms effect, Reference 211
- Appendix A.3., Table A.3., Polymer PLA; Microorganisms effect, Reference 213
- Appendix A.3., Table A.3., Polymer PLA; Microorganisms effect, Reference 214
- Appendix A.3., Table A.3., Polymer PLA; Microorganisms effect, Reference 218
- Appendix A.3., Table A.3., Polymer PLA; Microorganisms effect, Reference 219
- Appendix A.3., Table A.3., Polymer PHA; Temperature effect, Reference 258
- Appendix A.3., Table A.3., Polymer PHA; Seawater effect, Reference 262
- Appendix A.3., Table A.3., Polymer PHA; UV effect, Reference 230 – entry 2
- Appendix A.3., Table A.3., Polymer PHA; Microorganisms effect, Reference 234
- Appendix A.3., Table A.3., Polymer TPS; Water effect, Reference 236
- Appendix A.3., Table A.3., Polymer TPS; Water effect, Reference 237
- Appendix A.3., Table A.3., Polymer TPS; Microorganisms effect, Reference 243
- Appendix A.3., Table A.3., Polymer TPS; Microorganisms effect, Reference 245
- Do all the characterizing methods have a respective predictive model? Reviewing the paper seems like most of the characterizing methods are based on experimental methods.
R: The predictive models that exist for environmental fatigue and creep are based on experimental evidence and are, thus, mostly fitted a posteriori – albeit the amount of relevant data from which a comprehensive/general model can be derived is still scarce. Similarly, for static scenarios not related to fatigue nor creep, predictive models to estimate the influence of a different mechanism on the properties of a given polymer matrices are also based on experimental evidence. Exceptionally, for some polymers, the water uptake mechanism may typically follow a Fickean evolution law (or a modified version of it), but this evolution does not necessarily correlate with fatigue or creep performance, the aim of this review.
- The paper reviews the physical and chemical phenomena causing fatigue and creep, is surface pitting due to impurities considered one of the physical phenomena?
R: We thank the reviewer for the comment. This review is exclusively focused on the impact of environmental factors (water uptake, UV radiation, lightning strike, etc.) on the creep and fatigue of polymer-matrix composites; no characterisation on the influence of the material impurities or surface defects on creep and fatigue when exposed to the mentioned environmental factors is provided by the reviewed literature.
- Temperature is discussed in the paper lots of times, but it is not clear how the temperature change in the real world like a drastic change of temperature from hot to cold or vice versa would have effect on any of the polymers, matrix, and composites.
R: Thank you very much for the suggestion. In the temperature section (Part I, section 5), we agree with the reviewer’s comment and corrected the text accordingly: “Also, the higher the order of thermal shocks, i.e., greater and/or faster temperature variations, the higher is the probability of these effects to happen, as well as their severity” (lines 447-449).
- Line 450; Could this heat-absorbing problem be tackled by adding electric conductive materials into the polymer? The electric current will be transported to storage when the lighting strikes the PMC.
R: Suggestion accepted. Yes, although the problem cannot be currently fully solved, a more electrically conductive matrix can lead to better performance, i.e., lesser damage depth. This can be done, by techniques such as either modifying the resin with electrically conductive nanoparticles or by using conductive polymers as reinforcement. However, manufacturing constraints and/or lack of maturity of these technologies make their implementation difficult. In light of this, in the Lightning Strike section (Part I, section 6) we have revised the text accordingly: “Despite some solutions that have been proposed to mitigate the problem, such as the inclusion of electrically conductive nanoparticles in the polymer or the use of better conductive matrices, their maturity and ease of manufacturing are still the limiting factor that hampers their wide-scale implementation.” (lines 471-474).
Round 2
Reviewer 3 Report
Thank you for your good work. I recommend to accept the article for publication in its present form.